



# Features of mid- and high-latitude low-level clouds and their relation to strong aerosol effects in the Energy Exascale Earth System Model version 2 (E3SMv2)

Hui Wan[1], Abhishek Yenpure[2], Berk Geveci[2], Richard C. Easter[1], Philip J. Rasch[3], Kai Zhang[1], and Xubin Zeng[4]

[1]Atmospheric, Climate, and Earth Sciences Division, Pacific Northwest National Laboratory, Richland, WA, USA
[2]Kitware Inc., Clifton Park, NY, USA
[3]Department of Atmospheric Sciences, University of Washington, Seattle, WA, USA
[4]Department of Hydrology and Atmospheric Sciences, University of Arizona, Tucson, Arizona, USA

**Correspondence:** Hui Wan (Hui.Wan@pnnl.gov)

**Abstract.**

The E3SMv2 model, like various other global climate models that include representations of aerosol-cloud interactions, uses an empirically chosen lower bound on the simulated in-cloud cloud droplet number concentration (CDNC) to help constrain the effective radiative forcing of anthropogenic aerosols, $ERF_{aer}$. This study identifies where ultra-low CDNCs (i.e., concentrations lower than $10 \, \mathrm{cm^{-3}}$) occur in the stratiform and shallow convective clouds simulated by E3SMv2 and which of the occurrences have the strongest impact on $ERF_{aer}$. Process-level analyses are presented to reveal characteristics of the cloud droplet formation and removal processes associated with impactful ultra-low CDNCs.

Simulations performed with present-day emissions show that ultra-low CDNCs are most frequently found over the mid- and high-latitude oceans in both hemispheres, while the occurrences are also frequent in polluted continental regions despite the high aerosol concentrations. Ultra-low CDNCs with the largest impacts on the simulated regional and global mean $ERF_{aer}$ are found in the lower troposphere in the Northern Hemisphere middle and high latitudes. These cases are typically associated with large cloud fractions, strong water vapor condensation, weak turbulence, and lack of cloud droplet nucleation from aerosol activation. Under such atmospheric conditions, boosting aerosol activation and enhancing turbulent mixing of cloud droplet number can increase the simulated CDNCs, although the magnitude of the global mean $ERF_{aer}$ increases undesirably. The reason for this model behavior is discussed. Overall, our study suggests that mid- and high-latitude low-level stratus occurring under weak turbulence is a cloud regime worth further investigating for the purpose of identifying and addressing the root causes of ultra-low CDNCs and strong $ERF_{aer}$ in E3SM.

## 1 Introduction

The effects of anthropogenic aerosols on the energy budget of the Earth, especially the effects involving changes in cloud properties and cloud life cycle, continue to be a major challenge in our understanding of the Earth system and our ability to numerically model it (see, e.g., summaries in Seinfeld et al., 2016; Bellouin et al., 2020). The Energy Exascale Earth System





Model (E3SM, Leung et al., 2020, https://e3sm.org/) is a global numerical model developed for research and predictions related to climate change. In the publicly released E3SMv1 and v2 (Golaz et al., 2019, 2022), the global mean effective radiative forcing of anthropogenic aerosols, denoted hereafter by $\mathrm{ERF_{aer}}$, was $-1.65\ \mathrm{W\,m^{-1}}$ and $-1.52\ \mathrm{W\,m^{-1}}$, respectively,

when estimated using pre-industrial climate simulations (Golaz et al., 2022, see the end of Sect. 4.2 therein). Both values were deemed large in magnitude compared to the results from various other models (e.g., Smith et al., 2020; Zhang et al., 2022a) as well as the estimates based on satellite retrievals (e.g., Bellouin et al., 2020).

Zhang et al. (2022a) showed that the strong $\mathrm{ERF_{aer}}$ in E3SMv1 was attributable primarily to the aerosol indirect effect. Golaz et al. (2022) noted that, similar to the results reported for various other models, e.g., Lohmann et al. (2000), Hoose et al. (2009),

Wang and Penner (2009), and Neubauer et al. (2019), the magnitude of the aerosol indirect effect (and hence $\mathrm{ERF_{aer}}$) in E3SM could be pragmatically reduced by imposing a lower bound on the in-cloud Cloud Droplet Number Concentration (CDNC), i.e., by letting

$$\mathrm{CDNC}_b = \max\left(\mathrm{CDNC}, \mathrm{CDNC_{min}}\right), \tag{1}$$

where CDNC in Eq. (1) is the in-cloud droplet number concentration obtained by solving the model's prognostic and diagnostic

equations, $\mathrm{CDNC_{min}}$ is an empirically chosen constant on the order of 10 to a few tens of droplets per $\mathrm{cm}^3$ of air, and $\mathrm{CDNC}_b$ is the bounded value used in the calculation of cloud microphysics. (Since the atmosphere component of E3SM allows for partial cloudiness in a grid box—as is commonly done in global atmospheric models, we clarify that all CDNCs mentioned in this paper are the in-cloud values in the respective grid boxes. Whether the other physical quantities discussed below are in-cloud or grid box averages is clarified when the quantities are introduced.)

On the one hand, low CDNCs have been observed in the real world and examined by researchers, see the examples listed in Hoose et al. (2009) and more recent evidences from, e.g., Wood et al. (2018), O et al. (2018) and Grosvenor et al. (2018). On the other hand, lower bounds of CDNC not only were used in earlier simulations of aerosol indirect effects (see examples summarized in Hoose et al., 2009) but continue to be used in most recent models, e.g., $40\ \mathrm{cm}^{-3}$ in Salzmann et al. (2022), $20\ \mathrm{cm}^{-3}$ in van Noije et al. (2021), $10\text{–}20\ \mathrm{cm}^{-3}$ in Mignot et al. (2021), and $10\ \mathrm{cm}^{-3}$ or $40\ \mathrm{cm}^{-3}$ in Neubauer et al. (2019).

From a model developer's perspective, in order to understand the strong aerosol indirect effect in E3SM from its roots, it will be useful to understand where and why very low CDNCs occur in the simulations. To that end, the present paper combines sensitivity experiments with online and offline diagnostics to answer the "where" question about the occurrences of ultra-low CDNCs (i.e., concentrations lower than $10\ \mathrm{cm}^{-3}$) in E3SMv2 and provide some initial clues to the "why" question. Specifically, we identify the geographical locations and altitudes with frequent occurrences of ultra-low CDNCs and determine

which of those locations have the largest impacts on the simulated $\mathrm{ERF_{aer}}$. The features of the atmospheric conditions as well as cloud droplet formation, transport, and removal processes associated with ultra-low CDNCs are also discussed.

The remainder of the paper is organized as follows. Section 2 provides a description of E3SMv2's atmosphere component, focusing on the cloud and aerosol process parameterizations most relevant to this study. Section 3 presents an overview of the numerical experiments. Section 4 identifies where ultra-low CDNCs occur in E3SMv2 simulations and which of the occur-

rences have the strongest impact on $\mathrm{ERF_{aer}}$. Section 5 presents process-level analyses of the atmospheric conditions and cloud





droplet formation, transport, and removal processes associated with the simulated ultra-low CDNCs. Section 6 presents some additional sensitivity experiments and discussions. Section 7 summarizes the results and draws conclusions.

## 2 Model description

The atmosphere component of E3SMv2, commonly referred to as EAMv2 (see Sect. 2.1 in Golaz et al., 2022) is a global
hydrostatic atmospheric model including comprehensive representations of large- and small-scale fluid dynamics, clouds, radiation, and aerosol life cycles, as well as the interactions among these physical processes (Fig. 1a). Broadly speaking, the parameterizations used in EAMv2 divide the simulated clouds into two categories: (I) deep convection and (II) shallow convection and stratiform clouds. The two categories of clouds can co-exist in a model grid box, but the latter is where the CDNCs are bounded so as to reduce the magnitude of the global mean $ERF_{aer}$, and hence the focus of this study.
To provide necessary background information for the reasoning in this paper and facilitate the comprehension of our analyses, the remainder of this section includes an overview of the sources and sinks of cloud droplets in stratiform and shallow convective clouds in EAMv2 (Sect. 2.1), a brief summary of the representation of aerosol life cycles (Sect. 2.2), and a description of the parameterization that represents the nucleation, evaporation, and turbulent mixing of cloud droplet number (Sect. 2.3).

### 2.1  Sources and sinks of cloud droplets

For the startiform and shallow convective clouds, EAMv2 uses the parameterization of Gettelman and Morrison (2015), commonly referred to as MG2, to represent the cloud microphysics processes, i.e., collision-coalescence of cloud condensate and formation of precipitation, as well as gravitational settling of the hydrometeors. The parameterization uses a two-moment approach, in which the mass and number mixing ratios of cloud droplets are predicted with separate prognostic equations.
Among the atmospheric processes considered in EAMv2, those depicted with gray boxes in the schematic in Fig. 1a can directly affect cloud droplet mass or number in stratiform and shallow convective clouds. The respective tendencies (i.e., rates of change with respect to time) are shown in Fig. 1b for droplet mass and in Fig. 1c for droplet number, presented here as the monthly and annual mean vertical integrals averaged over the middle and high latitudes. (The low latitudes, 30°S to 30°N, are excluded because the deep convective clouds dominate in those regions.) In terms of such global-scale integrals, the MG2
cloud microphysics parameterization leads to net sinks of both the mass and the number of cloud droplets (Fig. 1b-c). The primary source of cloud droplet mass is the condensation of water vapor calculated by the parameterization named CLUBB (Cloud Layers Unified By Binormals, Larson, 2017; Larson and Golaz, 2005; Golaz et al., 2002; Larson et al., 2002), which uses higher-order closures for parameterizing turbulence. The primary source of cloud droplet number is droplet nucleation (Fig. 1c), which is represented with a comprehensive parameterization following Ghan and Easter (2006) and Easter et al.
(2004), as elaborated below in Sect. 2.3. Condensed water detrained from deep convection is also a source for droplet mass and number for stratiform and shallow convective clouds, but this source is small in the mean budget in middle and high latitudes (Fig. 1b-c). Transport by the resolved winds (i.e., large-scale advection) and by the parameterized updrafts and downdrafts in





deep convection can lead to local changes of cloud droplet mass and number, but the net effects after temporal and spatial averaging are very small (Fig. 1b-c).

Geographical distributions of the vertically integrated annual mean budget terms are shown in Fig. S1 in the Supplement. There, we see again that the impacts of resolved transport and parameterized deep convective transport are small. In the middle and high latitudes, the primary sources of cloud droplet mass and number are water vapor condensation and droplet nucleation, respectively, while cloud microphysics is the primary sink for both droplet mass and droplet number.

## 2.2   Aerosols

The abundance of aerosol particles in the atmosphere is simulated in EAMv2 by solving time evolution equations of particle mass and number mixing ratios. The wide ranges of particle size and composition are taken into account by considering seven chemical components (sulfate, black carbon, primary organic matter, secondary organic matter, marine organic matter, dust, sea salt) and using four lognormally distributed modes to represent the statistical distributions of particle size and composition at specific time instances and locations. Furthermore, aerosol particles in EAMv2 are divided into two sub-populations of

different attachment states: the interstitial aerosols are those found outside cloud droplets, while the cloud-borne aerosols are those embedded in cloud droplets. EAMv2 solves separate mass and number mixing ratio equations for the different compositions, modes, and attachment states listed above. Detailed descriptions of the aerosol life cycles in EAMv2 can be found in Wang et al. (2020), Liu et al. (2016), and Liu et al. (2012).

The activation of aerosol particles leads to the formation of new cloud droplets, a process commonly referred to as cloud

droplet nucleation. From the aerosol life cycle perspective, activation changes the attachment state of an aerosol particle from interstitial to cloud-borne, while the evaporation of cloud droplets can result in aerosol resuspension, i.e., the conversion of cloud-borne aerosols back to the interstitial state. Further details of the parameterization in EAMv2 involving aerosol activation can be found in Sect. 2.3 below.

## 2.3   Nucleation, evaporation, and turbulent mixing of cloud droplet number

In EAMv2, the changes in cloud droplet number mixing ratio caused by cloud droplet nucleation or evaporation is represented by a scheme based on Ghan and Easter (2006) and Easter et al. (2004), similar to what has been summarized in Sect. S1.1.8 of Liu et al. (2012). The following features of the implementation in EAMv2 (and also in EAMv1, Rasch et al., 2019) are important for understanding the results presented in later sections.

First, the nucleation, evaporation, and turbulent mixing of cloud droplet number are tightly coupled both physically and nu-

merically. For brevity, we refer to the parameterization in EAMv2 that describes this collection of processes as DROPMIXNUC following the Fortran subroutine name in the EAMv2 code. EAMv2 does not assume vertically constant CDNCs in multi-layer cloudy regions; rather, the parameterized turbulence is responsible for the upward transport of cloud droplets formed at cloud base and for the subgrid-scale mixing of droplets across vertical layers. This design choice is different from the assumptions used in the ECHAM4 model described in Sect. 2.3 of Lohmann et al. (1999) and in various more recent models such as

ECHAM-HAM (Zhang et al., 2012; Tegen et al., 2019; Neubauer et al., 2019), ECHAM-SALSA (Kokkola et al., 2018), and



UKESM1 (Grosvenor and Carslaw, 2020). The implementation in these other models can be interpreted as assuming turbulent transport is sufficiently strong to vertically homogenize CDNC in a contiguous cloudy region within the same time step of the numerical simulation.

Second, droplet nucleation in EAMv2 (and also v1) is treated differently at cloud base and inside newly formed clouds,
see Sect. 2.1 in Ghan and Easter (2006) as well as Sect. 2.3.2 versus Sect. 2.3.3 below. In both cases, however, the fractions of interstitial aerosol particles in different lognormal modes that are activated to form cloud droplets are calculated using the parameterization of Abdul-Razzak and Ghan (2000, hereafter ARG2000), assuming the supersaturation of water vapor that leads to aerosol activation is caused by the adiabatic ascents of air parcels. In EAMv2 (and also v1), the characteristic speed of such ascents, $w_{\mathrm{act}}^*$, is estimated as

$$w_{\mathrm{act}}^* \quad = \quad \max\left(w_{\mathrm{act,min}}, \sigma_w\right), \quad \text{where} \tag{2}$$

$$\sigma_w \quad = \quad \min\left(\sigma_{\max}, \sqrt{\overline{w'^2}}\right). \tag{3}$$

Here $\overline{w'^2}$ is the sub-grid variance of the vertical velocity of air calculated by CLUBB. The minimum ascent velocity, $w_{\mathrm{act,min}}$, is set to $0.1 \mathrm{\ m\,s}^{-1}$, and $\sigma_{\max} = 10 \mathrm{\ m\,s}^{-1}$.

Overall, the impact of the DROPMIXNUC parameterization on cloud droplet number can be understood as a grid box mean
droplet number tendency expressed as

$$\left(\frac{\partial \overline{N_l}}{\partial t}\right)_{\mathrm{DROPMIXNUC}} = \left(\frac{\partial \overline{N_l}}{\partial t}\right)_{\mathrm{mix}} + \left(\frac{\partial \overline{N_l}}{\partial t}\right)_{\mathrm{nuc\text{-}evap}}, \tag{4}$$

where $N_l$ denotes the droplet number mixing ratio, and an overline denotes the spatial average over a grid box. The first right-hand-side term in Eq. (4) is turbulent mixing parameterized using the eddy diffusivity theory. The second term is formulated as

$$\left(\frac{\partial \overline{N_l}}{\partial t}\right)_{\mathrm{nuc\text{-}evap}} \quad = \quad \left(\frac{\partial \overline{N_l}}{\partial t}\right)_{\mathrm{cld\text{-}regen}}$$

$$+ \left(\frac{\partial \overline{N_l}}{\partial t}\right)_{\mathrm{cld\text{-}shrink}} + \left(\frac{\partial \overline{N_l}}{\partial t}\right)_{\mathrm{cld\text{-}grow}}$$

$$+ \left(\frac{\partial \overline{N_l}}{\partial t}\right)_{\mathrm{cld\text{-}base\text{-}nuc}} + \left(\frac{\partial \overline{N_l}}{\partial t}\right)_{\mathrm{vert\text{-}detr}} . \tag{5}$$

The first term on the right-hand side of Eq. (5) represents droplet nucleation by cloud regeneration inside persisting clouds (see Sect. S1.1.8 in Liu et al., 2012) but is neglected in EAMv2 (and v1). The other four terms represent droplet evaporation
associated with cloud fraction decrease (Sect. 2.3.1 and Fig. 2a), droplet nucleation associated with cloud fraction increase (Sect. 2.3.2 and Fig. 2b), nucleation at cloud base (Sect. 2.3.3 and Fig. 2c), and the evaporation of droplets detrained by turbulence (Sect. 2.3.4 and Fig. 2d). The cloud fraction used in DROPMIXNUC is the liquid cloud fraction calculated by CLUBB.





### 2.3.1 Droplet evaporation associated with cloud fraction decrease

If the cloud fraction in a grid box has decreased from the previous time instance at which the DROPMIXNUC parameterization was calculated, the decrement in cloud fraction is assumed to correspond to disappeared clouds (Fig. 2a). The decrement in the grid box mean droplet number mixing is assumed to be proportional to (I) the change in cloud fraction and (II) the in-cloud mean droplet number concentration of the previous time instance, namely,

$$\left(\frac{\partial \overline{N_l}}{\partial t}\right)_{\text{cld-shrink}} \Delta t = \widehat{N_l}(t)\Big[f_c(t+\Delta t) - f_c(t)\Big],\tag{6}$$

where $\Delta t$ is the time step size used by the parameterization, $\widehat{N_l}(t)$ is the in-cloud mean droplet number mixing ratio at time instance $t$, while $f_c(t)$ and $f_c(t+\Delta t)$ are cloud fractions at the old and new time instances, respectively. As a result of this evaporation, the previously cloud-borne aerosol particles are resuspended, resulting in increases in the grid box mean interstitial aerosol number and mass concentrations.

### 2.3.2 Droplet nucleation associated with cloud fraction increase

If the cloud fraction in a grid box has increased from the previous time instance, the increment is assumed to correspond to newly formed clouds (Fig. 2b). For these new clouds, the ARG2000 parameterization is used to calculate the fractions of activated aerosol particles, denoted below as $f_{a,i}, i = 1, \ldots, M$, with M being the total number of aerosol modes. Note that $f_{a,i}$ depends critically on $w^*_{\text{act}}$ as well as on the aerosol properties. These fractions are applied to the mean interstitial aerosol number mixing ratios in the grid box, denoted below by $\overline{N}_{a,i}, i = 1, \ldots, M$. Each activated aerosol particle is assumed to

correspond to one newly nucleated cloud droplet. This leads to an increase in grid box mean droplet number mixing ratio that is proportional to the cloud fraction increment, namely,

$$\left(\frac{\partial \overline{N_l}}{\partial t}\right)_{\text{cld-grow}} \Delta t = \Big[f_c(t+\Delta t) - f_c(t)\Big]\sum_{i=1}^{M} f_{a,i}(t)\,\overline{N}_{a,i}(t).\tag{7}$$

### 2.3.3 Droplet nucleation at cloud base

A grid box is assumed to have a fraction of its horizontal area covered by cloud base if the cloud fraction in the current grid box is larger than the cloud fraction in the grid box below by at least 1% (Fig. 2c). The cloud fraction difference between the two layers (a positive number) is defined as the cloud base fraction, $f_b$, of the grid box. At cloud base, the fractions of aerosol particles being activated from the below-base layer are calculated using the parameterization of ARG2000. These activation fractions are applied to the turbulent fluxes of interstitial aerosols across a unit area at the cloud base, $\overline{\mathcal{F}}_{a,i}$, where

$\overline{\mathcal{F}}_{a,i} = w^*_{\text{act}}\,\overline{N}_{a,i}, \quad i = 1, \ldots, M.$            (8)





Each activated aerosol particle is assumed to correspond to one newly nucleated cloud droplet. The resulting grid box mean droplet number tendency is

$$
\begin{aligned}
\left(\frac{\partial \overline{N_l}}{\partial t}\right)_{\text{cld-base-nuc}} &= \frac{f_{\text{b}}}{\Delta z} \sum_{i=1}^{M} f_{a,i}\, \overline{\mathcal{F}}_{a,i} \\
&= \frac{f_{\text{b}}}{\Delta z} \sum_{i=1}^{M} f_{a,i}\, w^{*}_{\text{act}}\, \overline{N}_{a,i},
\end{aligned}
\tag{9}
$$

where $z$ is geopotential height and $\Delta z$ is the thickness of the model layer. The aerosol number concentrations $\overline{N}_{a,i}$ in Eqs. (8) and (9) are the concentrations in the model layer right below cloud base.

The droplet number increment expressed in Eq. (9) and the turbulent mixing of cloud droplets (i.e., the first right-hand-side term in Eq. 4) are numerically solved together using an explicit time-stepping method with adaptive step sizes, where the $\Delta t$ in Eqs. (6) and (7) is subdivided into smaller time steps to fulfill the stability requirement.

To aid in the interpretation of some of the results presented later in Sect. 5.2, it is worth clarifying that when cloud fraction is non-zero in contiguous layers in the same grid column, cloud fraction may vary in the vertical, and hence cloud base may be identified in multiple layers according to the criterion stated at the beginning of Sect. 2.3.3. In this paper, we refer to the cloud base identified in a grid box as the *local cloud base* of that grid box. If the model layer closest to the Earth's surface is cloud-free, then the interface between the lowest model layer with non-zero cloud fraction and the layer below can be referred 190    to as, in a loose sense, the main cloud base (or lowest cloud base) in the grid column. The DROPMIXNUC parameterization does not distinguish these two types of cloud base. Both of them lead to source of cloud droplet number expressed by Eq. (9).

Given the definition of local cloud base noted above, one can imagine if a grid column has constant cloud fractions across multiple layers, then these layers will have only a main base and no local base in other cloudy layers. In such a scenario, considering that droplet nucleation by cloud regeneration inside persisting clouds (i.e., the first right-hand-side term in Eq. 5) 195    is neglected, if the cloud fractions do not increase with time, there will be no droplet nucleation above the main cloud base, and turbulent transport will be the only possible source of cloud droplet number in the cloudy layers away from the main base. This design feature is important to keep in mind for some of the discussions in Sects. 5.2.

### 2.3.4 Droplet evaporation after turbulent transport into clear air

When cloud droplets are transported by turbulence from a layer with a larger cloud fraction into a layer with a smaller cloud, a 200    fraction of the droplets are evaporated based on the difference in the two cloud fractions, and the cloud-borne aerosols therein are resuspended (Fig. 2d). This is done for transport to both the layer above and the layer below.

As a side note for the readers working with EAMv2, we emphasize here that the turbulent mixing of cloud droplet number (and all aerosol species) is numerically solved inside the DROPMIXNUC parameterization and is not handled by CLUBB.



## 2.4 Implementation of CDNC$_{\text{min}}$ in EAMv2

CDNC$_{\text{min}}$, the lower bound of CDNC, is set to $10 \text{ cm}^{-3}$ in the default EAMv2. Because the model's time integration uses sequential splitting to numerically couple most of the parameterized processes, the value of CDNC in a grid box, if diagnosed, would change multiple times within the 30-min time step depicted in Fig. 1a as well as during each of the 5-min sub-cycles used for CLUBB, DROPMIXNUC, and MG2. For our discussions here, it is important to clarify that the lower bound is applied after the DROPMIXNUC parameterization has been calculated and the resulting tendency of cloud droplet number has been applied, and before the cloud microphysical processes are calculated, see boxes (III), (*), and (IV) in Fig. 1a.

Due to the above-mentioned sequence of calculations, the lower bound, CDNC$_{\text{min}}$, directly affects all cloud microphysics processes in the MG2 cloud microphysics parameterization that depend on cloud droplet number concentration. In addition, CDNC$_{\text{min}}$ indirectly affects the parameters of the cloud droplet size distribution that are diagnosed near the end of MG2 (box IV in Fig. 1a) and passed subsequently to the radiation parameterization for the calculation of cloud optical properties (box V in Fig. 1a).

## 3 Numerical experiments and analysis methods

As mentioned in the introduction, this study aims at obtaining a process-level characterization of ultra-low CDNCs in stratiform and shallow convective clouds simulated by EAMv2. Our overall strategy is to first identify the geographical locations where ultra-low CDNCs occur most frequently as well as the locations where the estimated ERF$_{\text{aer}}$ is most sensitive to CDNC$_{\text{min}}$ (Sect. 4). We then examine characteristics of the cloud formation processes associated with ultra-low CDNCs (Sect. 5). Additional sensitivity experiments are used to assist further reasoning about ultra-low CDNCs and their relation to ERF$_{\text{aer}}$ (Sect. 6). Two types of simulations are presented in this paper, as explained below.

### 3.1 Nudged simulations for estimating ERF$_{\text{aer}}$

In this study, ERF$_{\text{aer}}$ is estimated following the methodology used by the Aerosol Comparisons between Observations and Models (AeroCom) intercomparision activities (see, e.g., Gliß et al., 2021; Myhre et al., 2013). Simulations were conducted with interactive atmosphere and land surface models. The sea surface temperature and sea ice extent were prescribed using climatological values. Pairs of otherwise identical simulations were conducted using the present-day (PD) or pre-industrial (PI) emissions of aerosols and precursors, with the PD emissions set to the 2005–2014 averages and the PI emissions set to values representing the year 1850. ERF$_{\text{aer}}$ was calculated as the PD–PI difference in the net radiative flux at the top of the model atmosphere.

The horizontal winds in these pairs of PD–PI simulations were nudged to the ERA-Interim reanalysis (Dee et al., 2011) to help distinguish signals of aerosols effects from noise caused by natural variability (Zhang et al., 2022b; Sun et al., 2019; Zhang et al., 2014; Kooperman et al., 2012). Zhang et al. (2022a) showed that 1-year nudged E3SM simulations were sufficient for revealing key signals in the annually averaged global mean, zonal mean, and global patterns of ERF$_{\text{aer}}$. In this study,





however, the sensitivity of $ERF_{aer}$ to $CDNC_{min}$ (i.e., $\Delta ERF_{aer}$ caused by $\Delta CDNC_{min}$) is examined also in different seasons (see Fig. 4), which requires more simulation years to distinguish signal from noise. Hence we chose to conduct one pair (PD and PI) of 10-year nudged simulations from 2005 to 2014 using the default EAMv2 (i.e., configuration nc10 in Table 1, with $CDNC_{min} = 10 \text{ cm}^{-3}$) and a second pair with $CDNC_{min}$ set to 0 (i.e., configuration nc00 in Table 1).

Inspired by the results described in Sects. 4 and 5, we conducted several additional pairs of PD–PI simulations. Group
II listed in Table 1 had $CDNC_{min}$ set to 10 $\text{cm}^{-3}$ but only in the lower troposphere (between ∼600 hPa and the Earth's surface, configuration nc10_600hPa) or only in grid boxes with cloud fraction higher than 0.9 (configuration nc10_f0.9). The experiments in Group III used no lower bound, but either the characteristic updraft velocity used in the droplet nucleation parameterization was artificially enhanced by a factor of 10 (configuration nc00_w10), or both the characteristic updraft and the turbulent mixing coefficient were enhanced by a factor of 10 (nc00_w10k10, Table 1). The group-II and group-III experiments
were used for assessing the model's responses in the annual mean $ERF_{aer}$, hence these simulations were integrated for one year after three months of spin-up.

### 3.2 Free-running simulations for characterizing ultra-low CDNCs and the associated atmospheric conditions

To carry out process-level analysis of the occurrences of ultra-low CDNCs in EAM, we present a 10-year free-running simulations conducted under forcing conditions of the year 2010 and with the lower bound $CDNC_{min}$ turned off (i.e., configuration
nc00). The reason for not using nudging in this simulation was that free-running simulations were more convenient to set up and would reflect the model's original behavior. That said, we do not expect nudging horizontal wind will substantially modify the cloud features discussed here.

Our analyses of the free-running simulation relied heavily on the online budget analysis and conditional sampling capabilities developed for EAM by Wan et al. (2022) as well as offline interactive data exploration enabled by the visualization application
ParaView. During a simulation, CDNCs were diagnosed at the point in the time loop where ultra-low values would be bounded for the first time in each 30-min "main" model time step when the lower bound was in use, as depicted by the dashed box marked with "(*)" in Fig. 1a. CDNC values in the range of 0.01–9.99 $\text{cm}^{-3}$ are referred to as "ultra-low" in this paper. We note that this range excludes the cases with CDNC = 0, as the zeros typically correspond to cloud-free conditions and are not the focus of this study. For comparison with the ultra-low cases, CDNC values of 30–50 $\text{cm}^{-3}$ and higher than 50 $\text{cm}^{-3}$ were
also sampled and are discussed in Sect. 5.

### 4 Ultra-low CDNCs and $ERF_{aer}$

This section aims to assess the frequencies and locations of the occurrences of ultra-low CDNCs in stratiform and shallow convective clouds simulated by EAMv2, along with identifying the locations at which the simulated $ERF_{aer}$ is most sensitive to the lower bound, $CDNC_{min}$.



## 4.1 Sensitivity of ERF$_{aer}$ to CDNC$_{min}$

We first compare the ERF$_{aer}$ simulated by EAMv2 with or without using the lower bound. The 10-year mean global and annual mean ERF$_{aer}$ is $-1.71\ \mathrm{W\,m^{-2}}$ in the nc00 configuration and $-1.43\ \mathrm{W\,m^{-2}}$ in nc10 (Fig. 3a). Figure 4a shows a global map of the annual mean ERF$_{aer}$ derived from the nc00 configuration (i.e., no lower bound). Figures 4b and 4c show the signed absolute and relative changes, respectively, caused by imposing a CDNC$_{min}$ of 10 cm$^{-3}$. In all three panels, masked out in white are the geographical locations with statistically small ERF$_{aer}$ in nc00, i.e., where an absolute value of the 10-year annual mean ERF$_{aer}$ is smaller than the estimated two standard deviations of the 10-year annual mean.

A comparison between Fig. 4a and Fig. 4b suggests the lower bound reduces ERF$_{aer}$ globally except over the major deserts. In the middle latitudes of both hemispheres, the spatial pattern of the change (Fig. 4b) largely matches the spatial pattern of ERF$_{aer}$ itself (Fig. 4a), meaning there is a rough proportionality between the magnitudes of change and the baseline values.

In the Arctic, however, the annual mean absolute changes are large, often exceeding 1 W m$^{-2}$ (Fig. 4b), while the baseline values are moderate (within $-2$ W m$^{-2}$ in general, Fig. 4a). Therefore, this region features the strongest relative reduction in the magnitude of ERF$_{aer}$, on the order or 50% to 100%, when a CDNC$_{min}$ of 10 cm$^{-3}$ is imposed (Fig. 4c). This strong sensitivity in the annual mean ERF$_{aer}$ is dominated by the changes in boreal summer, especially June and July, which can be seen in the zonally averaged monthly mean ERF$_{aer}$ differences shown in Fig. 4f. In the Northern Hemisphere (NH) middle latitudes, the 10-year mean monthly zonal mean changes are typically within 1.5 W m$^{-2}$, while near the North Pole, the changes are 5 W m$^{-2}$ or larger in June and July.

As can be expected from the results reported in Zhang et al. (2022a), the changes in ERF$_{aer}$ seen in Fig. 4 are dominated by the changes in the shortwave ERF$_{aer}$ (see Fig. S2 in the Supplement), which, in turn, is attributable primarily to the changes in the shortwave component of the aerosol indirect effect (see Fig. S3 in the Supplement).

## 4.2 Geographical distribution of ultra-low CDNCs

Figure 5a presents a global map of the 10-year mean annually averaged fraction of cloudy time steps and model layers that exhibit ultra-low CDNCs in the nc00 simulation. Here, "cloudy" is defined as $\overline{q_l} \geqslant q_{l,\min} = 10^{-15}\ \mathrm{g\,kg^{-1}}$, as is done at the point in the model's time loop where CDNC$_{min}$ is applied, and $\overline{q_l}$ is the instantaneous grid box mean cloud liquid mass mixing ratio in stratiform and shallow convective clouds. All vertical layers (72 in total in EAMv2) were included in the calculation of the fraction. Over the mid- and high-latitude oceans in both hemispheres, 25% to 55% of the cloudy time steps and vertical levels are found to have ultra-low CDNCs, and we see trends of higher percentages at higher latitudes.

The fraction shown in Fig. 5a can be interpreted as a conditional probability, $P\left(\mathrm{CDNC\ ultra\text{-}low}\,\middle|\,\overline{q_l} \geqslant q_{l,\min}\right)$, and was diagnosed using

$$P\left(\mathrm{CDNC\ ultra\text{-}low}\,\middle|\,\overline{q_l} \geqslant q_{l,\min}\right) =$$
$$P\left(\mathrm{CDNC\ ultra\text{-}low} \cap \overline{q_l} \geqslant q_{l,\min}\right) \big/ P\left(\overline{q_l} \geqslant q_{l,\min}\right). \tag{10}$$





To avoid drawing undue attention to large values of the conditional probability resulted from singularities associated with very small values of the denominator $P(\overline{q_l} \geqslant q_{l,\min})$, Fig. 5d shows $P(\overline{q_l} \geqslant q_{l,\min})$, namely the average fraction of cloudy cases among all model time steps and all vertical levels. Compared to the low latitudes and most continental areas, the mid- and high-latitude oceans are associated with more frequent presence of cloud liquid mass. Hence, the high frequencies of ultra-low
CDNC depicted by Fig. 5a over the mid- and high-latitude oceans are meaningful signals. The geographical distribution of the joint probability $P(\text{CDNC ultra-low} \cap \overline{q_l} \geqslant q_{l,\min})$ can be found in Fig. S4a in the Supplement.

One might also wonder whether the high frequencies seen in Fig. 5a and Fig. 5d have resulted from using a very low threshold of $q_{l,\min} = 10^{-15}\ \mathrm{g\,kg^{-1}}$ to define "cloudy" (i.e., cloud liquid present). Since $\overline{q_l}$ is the grid box mean value and the grid boxes are large (approximately $200\ \mathrm{km} \times 200\ \mathrm{km}$), and since numerical discretization and floating-point calculations both
introduce error, we can imagine a simulation producing $\overline{q_l}$ values that exceed $10^{-15}\ \mathrm{g\,kg^{-1}}$ but are nevertheless very small from a practical perspective and hence unlikely to have large impacts on the mean climate. To avoid drawing undue attention to such cases, we diagnosed, in the same nc00 simulation, fractional frequencies similar to those shown in Fig. 5a and Fig. 5d but with $q_{l,\min}$ set to $10^{-5}\ \mathrm{g\,kg^{-1}}$ in conditional sampling. The resulting global maps are presented in the middle column of Fig. 5, panels (b) and (e), which turn out to closely assemble the results obtained using the smaller threshold of $10^{-15}\ \mathrm{g\,kg^{-1}}$.
Further increasing $q_{l,\min}$ to $0.02\ \mathrm{g\,kg^{-1}}$ leads to substantial decreases in the count of cloudy cases (Fig. 5f versus Fig. 5e), but the mid- and high-latitude oceans still stand out as the regions with most frequent cloud liquid presence (Fig. 5f). Overall, the three sets of diagnostics presented in the figure suggest that over the middle and high latitude oceans, 25% to 55% of the cases (i.e., grid boxes and time steps) with cloud liquid presence are associated with ultra-low CDNCs. The feature is robust across a wide range of $q_{l,\min}$ used for defining cloud liquid presence.

Between the two hemispheres, the Southern Hemisphere (SH) shows slightly more frequent occurrences of ultra-low CDNCs (Fig. 5, upper row and Fig. S4, upper row). The absolute changes in ERF$_\text{aer}$ caused by imposing CDNC$_\text{min}$, however, are substantially smaller in SH than in NH (Fig. 4b). This is not surprising, given the significantly lower aerosol concentrations and their PD–PI differences in SH. Since this study was originally motivated by ERF$_\text{aer}$ in EAMv2, the analyses in the remainder of the paper will focus on NH.

In southeast (SE) China and SE United States, about 15% to 25% of the time steps and vertical levels with cloud liquid presence are associated with ultra-low CDNCs regardless of the value of $q_{l,\min}$ used in conditional sampling (Fig. 5, upper row). These percentages might seem unexpectedly high for continental regions with generally high aerosol concentrations. In the following, these two continental regions are further analyzed together with the Arctic region and the North Pacific storm track. The latter two regions were chosen because they feature both high frequencies of occurrence of ultra-low CDNCs (Fig. 5, upper row) and strong sensitivities of ERF$_\text{aer}$ to CDNC$_\text{min}$ (Fig. 4b).

## 4.3 Vertical distribution of ultra-low CDNCs

Based on the findings in Sects. 4.1 and 4.2, we selected four focus regions, namely the North Pacific storm track (oceanic areas in 30–60°N, 120–240°E), the Arctic region (all geographical locations northward of 66.5°N), SE China (land areas in 25–35°N, 110–120°E), and SE United States (land areas in 30–40°N, 270–280°E), to examine the vertical distribution of



the ultra-low CDNCs. Similar to Fig. 5, the upper row of Fig. 6 shows the 10-year mean fractional frequency of occurrences of ultra-low CDNCs in samples with cloud liquid presence, i.e., $P\left(\text{CDNC ultra-low}\,\middle|\,\overline{q_l} \geqslant q_{l,\min}\right)$, and the lower row shows the fraction of samples featuring $\overline{q_l} \geqslant q_{l,\min}$ among all time steps and grid boxes in the region and vertical layer, i.e., $P\left(\overline{q_l} \geqslant q_{l,\min}\right)$. The vertical profiles of the joint probability $P\left(\text{CDNC ultra-low} \cap \overline{q_l} \geqslant q_{l,\min}\right)$ can be found in Fig. S5 in the Supplement. The four columns in Fig. 6 show results for the four regions. The solid, dashed, and dotted lines in each panel are 10-year mean

annual averages corresponding to $q_{l,\min} = 10^{-15}$ g kg$^{-1}$, $10^{-5}$ g kg$^{-1}$, and 0.02 g kg$^{-1}$, respectively. Two standard deviations of 1-year averages are shown as color shading to indicate interannual variability.

The four regions differ in their frequencies of cloud liquid presence, as expected (Fig. 6, lower row), but similar features can be seen in the fractions of cloudy cases exhibiting ultra-low CDNCs (Fig. 6, upper row). In the upper troposphere (with pressure values lower than 300 hPa or 400 hPa, depending on the region), the occurrences of ultra-low CDNCs in liquid-

present cases vary substantially with $q_{l,\min}$, possibly due to the dominance of ice clouds and very small values of $P\left(\overline{q_l} \geqslant q_{l,\min}\right)$ at those altitudes. In the middle and lower troposphere (pressure > 400 hPa), the fractions of liquid-present cases associated with ultra-low CDNCs are rather insensitive to $q_{l,\min}$ (Fig. 6, upper row). Compared with the 10-year averages, the interannual variabilities are small, suggesting the mean frequencies shown here are statistically significant.

The clouds in EAMv2 in the upper troposphere are often cirrus, while those occurring between 600 hPa and the surface

are typically mix-phase or warm (liquid) clouds. To find out which of these cloud types play more important roles in causing the sensitivity of ERF$_{\text{aer}}$ to CDNC$_{\min}$, we conducted the sensitivity experiment nc10_600 hPa listed in Table 1, in which CDNC$_{\min} = 10$ cm$^{-3}$ was applied only to the model's vertical layers 48 to 72, corresponding to pressure values equal to or higher than about 600 hPa when the surface pressure is 1000 hPa. Compared to nc00 (no lower bound), bounding CDNC in all layers (i.e., nc10) reduces the magnitude of the global annual mean ERF$_{\text{aer}}$ by 0.34 W m$^{-2}$ in the year 2011 and by

0.19–0.36 W m$^{-2}$ in the 10-year period of 2005 to 2014 (Fig. 3b). Bounding CDNCs between 600 hPa and the surface gives a reduction of 0.33 W m$^{-2}$ in the year 2011, suggesting the ultra-low CDNCs in the lower troposphere play a dominant role in determining the sensitivity of ERF$_{\text{aer}}$ to CDNC$_{\min}$.

For the lower tropospheric layers between 600 hPa to about 900 hPa, Fig. 6b reveal that about 40% of the cloudy cases (i.e., cloudy time steps and grid boxes) in the Arctic region are associated with ultra-low CDNCs. Over the North Pacific storm track

(Fig. 6a), SE China (Fig. 6c), and SE USA (Fig. 6d), the typical percentages are between 20% and 40%. These are rather high annual mean percentages, especially for the polluted regions.

## 4.4   Cloud morphology

Having identified the locations of the most frequent occurrences of ultra-low CDNCs, we now start to examine the features of the model atmosphere in those cases, focusing on the lower troposphere and on the four regions discussed earlier in Fig. 6.

Our preliminary examination of the grid box mean cloud liquid mass mixing ratio ($\overline{q_l}$, unit: g kg$^{-1}$), the in-cloud droplet mass concentration (CDMC, unit: g cm$^{-3}$), and cloud fraction ($f_c$, unitless) suggested that the cases with ultra-low CDNCs showed moderate to high *composite mean* values for all three quantities. A subsequent investigation using *instantaneous* model output revealed the existence of two distinct regimes. In the upper row of Fig. 7, bivariate histograms of instantaneous $\overline{q_l}$ and



$f_c$ in the four focus regions are presented for the ultra-low CDNC cases sampled in the lower troposphere from 120 days
of 6-hourly instantaneous model output. The lower row of Fig. 7 shows bivariate histograms of CDMC and $f_c$. To include
different seasons, the 120 days used in this analysis consisted of 30 consecutive days per month from January, April, July, and
October of the first simulation year. The results from the four geographical regions turn out to show very similar features: the
majority of the samples with ultra-low CDNCs are associated with $f_c > 0.9$, typical $\overline{q_l}$ between 0.01 $\mathrm{g\,kg^{-1}}$ and 1 $\mathrm{g\,kg^{-1}}$,
and in-cloud droplet mass concentrations from 0.01 $\mathrm{g\,cm^{-3}}$ to 1 $\mathrm{g\,cm^{-3}}$. There is a second regime featuring very small cloud
fractions ($f_c < 0.1$), typical $\overline{q_l}$ lower than 0.01 $\mathrm{g\,kg^{-1}}$, and in-cloud droplet mass concentrations spanning a wide range from
about $10^{-5}$ $\mathrm{g\,cm^{-3}}$ to 1 $\mathrm{g\,cm^{-3}}$. Only a very small portion of the samples are associated with cloud fractions between 0.1 and
0.9.

These results naturally trigger the following questions:

1. Which of the two regimes plays a larger role in affecting the simulated global mean $\mathrm{ERF_{aer}}$?

2. Where do the samples in each regime occur in terms of cloud morphology? In other words, do the ultra-low CDNCs
   associated with small or large cloud fractions typically occur inside a cloudy region or at the boundaries between cloudy
   and cloud-free regions? Do they occur in small and scattered cloudy regions or are they associated with large (e.g.,
   synoptic-scale) cloud systems?

The answers to these questions may provide important clues for further investigations. For example, when ultra-low CDNCs are
associated with very small cloud fractions and are sporadically located (e.g., at the edges of cloudy regions), one can suspect
they might have been caused by numerical artifacts of spatial discretization. In contrast, if impactful ultra-low CDNCs are
found in large contiguous areas and at specific types of locations (e.g., in the center or at the bottom of synoptic-scale cloudy
regions), then the ultra-low values may reflect certain inherent features of the physical assumptions in the model.

Our sensitivity experiment nc10_f0.9 was designed to answer the first question. In this pair of PD and PI simulations, the
lower bound $\mathrm{CDNC_{min}}$ was applied only to grid boxes with cloud fractions higher than 0.9. The year-2011 mean global mean
$\mathrm{ERF_{aer}}$ change relative to nc00 amounts to a reduction of 0.22 $\mathrm{W\,m^{-2}}$ in magnitude, explaining a major portion (about 65%)
of the reduction of 0.34 $\mathrm{W\,m^{-2}}$ in magnitude obtained by eliminating all ultra-low CDNCs (Fig. 3b).

The second question asked above is not straightforward to answer using numerical metrics, so we chose to resort to visual
inspection using ParaView. Given the finding in Sect. 4.1 that $\mathrm{ERF_{aer}}$ simulated by EAMv2 features the strongest sensitivity
to $\mathrm{CDNC_{min}}$ in Arctic summer, Fig. 8 presents various vertical cross-sections in the Arctic circle along the prime meridian,
captured at 00 UTC on July 3 of the first simulation year. The two rows in the figure present the same quantities in the format of
color shading: cloud fraction in the left column, CDNC in the middle column, and CDMC in the right column. The quadrilateral
frames in the upper row of Fig. 8 indicate grid boxes with ultra-low CDNCs and small cloud fractions ($f_c < 0.1$). These cases
are found sporadically (Fig. 8a) and are associated with low in-cloud droplet mass concentrations (Fig. 8f). This combination
of features seems physically plausible. The quadrilateral frames in the lower row of Fig. 8 indicate EAMv2's grid boxes with
ultra-low CDNCs and large cloud fractions ($f_c > 0.9$). These grid boxes are located in the main body of a multi-layer cloud
system (Fig. 8d) where CDNCs are systematically low across a number of vertical layers and grid columns (Fig. 8e). The in-





cloud droplet mass concentrations in these grid boxes, however, can be moderate or high (e.g., more than $0.05\ \mathrm{g\,m^{-3}}$, Fig. 8f), which is consistent with the bivariate histogram shown earlier in the lower row of Fig. 7. This combination of features, i.e.,

ultra-low CDNC but high cloud fraction and substantial in-cloud droplet mass concentration, is counterintuitive. While Fig. 8 shows only one time instance, the distinct features of the two types of ultra-low CDNC cases are frequently seen in the 30 days of instantaneous output we examined from June 28 to July 27 of the first simulation year. An animation showing vertical cross-sections for the 30-day period can be found in the video supplement.

A similar analysis of cloud morphology has been carried out for the SE China region. To ease the search for relevant cases, we

examined the monthly mean regionally averaged vertical profiles of $P\left(\text{CDNC ultra-low} \mid \overline{q_l} \geqslant q_{l,\min}\right)$, $P\left(\text{CDNC ultra-low} \cap \overline{q_l} \geqslant q_{l,\min}\right)$, and $P\left(\overline{q_l} \geqslant q_{l,\min}\right)$ in the first simulation year and selected March as the focus month, as all three fractions were relatively high in that month (see third column of Fig. S6 in the Supplement). We then selected a model output time, namely 00 UTC on March 8, when the region of 110–120°E, 25–35°N was fully covered by low-level clouds judged from the vertically integrated low-level cloud fraction (Fig. 9a). In panels (b) and (c) of Fig. 9, cloud fractions in the region of 110–120°E, 25–35°N are

shown in semi-transparent color shading, and the grid boxes with ultra-low CDNCs and large (> 0.9, Fig. 9b) or small (< 0.1, Fig. 9c) cloud fractions are shown as light gray or dark blue solid 3D boxes, respectively, with the colors indicating the cloud fraction values. The two panels, Figs. 9b and 9c, reveal a feature similar to what has been seen for the Arctic: ultra-low CD-NCs associated with large cloud fractions are found in contiguous grid boxes inside large cloudy regions, while the ultra-low CDNCs associated with small cloud fractions are scattered in space. An animation showing the same kind of 3D visualization

for the 30-day period from February 28 to March 29 of the first simulation year can be found in the video supplement, which suggests the distinct features of the two types of ultra-low CDNC cases are typical over SE China.

Since the ultra-low CDNCs associated with large cloud fractions have a major impact on the global mean $\mathrm{ERF_{aer}}$ and are intriguingly located inside large cloudy regions, we focus on this category of cases in the next section, using composite analysis to understand characteristics of the droplet formation and removal processes.

# 5  Cloud droplet budget analyses

Generally speaking, ultra-low CDNCs in stratiform and shallow convective clouds could be caused by weak sources or strong sinks of droplet number, or both. To help identify the culprits, we present in Sect. 5.1 a budget analysis for the main groups of droplet formation, transport, and removal processes considered in EAMv2, namely the dynamical core and parameterizations depicted by gray boxes in Fig. 1a and described in Sect. 2.1. After that, Sect. 5.2 zooms into the DROPMIXNUC

parameterization to analyze the processes depicted in Fig. 2 and described in Sect. 2.3.

To put the numbers associated with ultra-low CDNCs into context, two additional CDNC ranges, 30–50 $\mathrm{cm^{-3}}$ and higher than 50 $\mathrm{cm^{-3}}$, were sampled for comparison, all under the condition of cloud fraction being higher than 0.9. For clarification, we note that the conditional sampling was done pointwise (i.e., gridboxwise rather than columnwise), as we intended to examine the local conditions associated with different CDNC ranges. At any time step, if one grid column had, for example,

two grid boxes with ultra-low CDNCs, and another grid column had ten grid boxes with ultra-low CDNCs, these grid boxes





would each contribute to the count of ultra-low CDNCs in their respective model layers, and the atmospheric conditions (or process rates) in those grid boxes were recorded to diagnose composite mean values.

The fractional frequency of occurrence of the three CDNC ranges with respect to all model time steps in the month of year and all grid boxes in the model layer and region is shown in Fig. 10a for Arctic July as a function of altitude (pressure). The

marks and lines indicate 10-year averages of the focus month. Color shading indicates two standard deviations of the July averages of 10 different years. Figure 10a shows that in Arctic July, ultra-low CDNCs occur significantly more often than higher concentrations, with a clear trend of higher frequencies of ultra-low concentrations towards the surface. In the figures presented in the remainder of this section, composite mean vertical profiles of various physical quantities are shown. (The results are masked out, i.e., not shown for the specific altitude or sampling condition, if the fractional frequency of occurrence

of the condition is lower than 0.1% in the focus month in any of the 10 years, to avoid showing results for very small samples.)

The composite mean cloud fractions of the sampled CDNC ranges are shown in Fig. 10e for Arctic July. The mean values are generally higher than 0.95 for all three CDNC ranges and are about 0.99 or higher for the ultra-low CDNCs. Due to these very high cloud fractions, although all process rates (and other physical quantities) shown in the remainder of this section are grid box mean values, we expect the in-cloud mean values to be similar.

## 5.1    Source problem or sink problem?

Panels (b), (c) and (d) in Fig. 10 present the composite mean cloud droplet number tendencies in Arctic July caused by the resolved transport, DROPMIXNUC, and cloud microphysics. The parameterized vertical transport by updrafts and downdrafts in deep convection and the detrainment from deep convection have negligible magnitudes and hence are not shown. This budget analysis shows that, overall, DROPMIXNUC is the primary source of cloud droplet number in this region and month of year,

while cloud microphysics and resolved transport are both significant sinks.

A comparison among the three CDNC ranges reveals that both the sources and the sinks associated with ultra-low CDNCs are markedly weaker (close to zero) than those seen for higher CDNCs (Fig. 10b–d). This feature is also seen in the other three regions (see Figs. S7–S9 in the Supplement), suggesting the ultra-low CDNCs are likely caused by weak sources rather than strong sinks.

## 5.2    Droplet nucleation and turbulence

Panels (f), (g) and (h) in Fig. 10 present a budget analysis similar to panels (b) to (d) but for the mass of cloud droplets. As expected from the construct of EAMv2 and its parameterizations, in most model layers in the lower troposphere (except for the very few layers closest to the Earth's surface), the turbulence and cloud parameterization CLUBB acts as the primary source of droplet mass while cloud microphysics is the primary sink. Somewhat counterintuitively, when comparing the three CDNC

ranges, we see the cases with ultra-low CDNCs are associated with substantially stronger mass sources and sinks (panels (f) to (h)) despite their much weaker number sources and sinks revealed by panels (b) to (d). This contrast in tendencies is consistent with the feature revealed earlier in Fig. 8 that the grid boxes with ultra-low CDNCs and large cloud fractions are





often associated with medium to high in-cloud droplet mass concentrations. Furthermore, the contrast between strong mass sources/sinks and weak number sources/sinks is also seen clearly in the other three regions, see Figs. S7–S9 in the Supplement.

To investigate why the droplet number sources associated with ultra-low CDNCs are weak while the corresponding droplet mass sources and strong, we decomposed the number tendency caused by the DROPMIXNUC parameterization, i.e., the quantity shown in Fig. 10c, into the various contributors described earlier in Sect. 2.3: droplet nucleation at the local cloud base in the grid box (Fig. 11a), droplet nucleation associated with cloud fraction increase in the grid box (Fig. 11b), and turbulent mixing (Fig. 11c). The evaporation associated with cloud fraction decrease and the evaporation of droplets detrained

by turbulence are both negligible in magnitude and hence not shown. The figure shows that the cases with higher CDNCs ($30$–$50 \, \mathrm{cm}^{-3}$ or $>50 \, \mathrm{cm}^{-3}$) are associated with significant droplet number sources due to nucleation at the local cloud base in the grid box (Fig. 11a); positive tendencies are seen in almost all layers in the lower troposphere, and peaks of nucleation rate are found in the near-surface layers (Fig. 11a, green and blue). The tendencies associated with turbulence mixing are negative (Fig. 11c, green and blue), suggesting droplets are transported away from where they are nucleated. Nucleation associated with

cloud fraction increase is negligible in most layers with pressure higher than about 800 hPa. The positive net droplet number tendencies seen earlier in Fig. 10c for the DROPMIXNUC parameterization in these higher CDNC ranges turn out to be small residuals between cloud-base nucleation (a substantial source) and turbulent mixing (a substantial sink). In contrast, in the cases with ultra-low CDNCs, the droplet number tendencies are very close to zero for all three processes (Fig. 11a-c, red) and for the total tendency caused by DROPMIXNUC (Fig. 10c, red).

Figure 11a reveals near-surface peaks of cloud-base droplet nucleation in the cases of higher CDNCs but negligible nucleation associated with ultra-low CDNCs. In Sect. 2.3.3, we have explained that this source of cloud droplet is associated with the turbulent influx of activated aerosols at the local cloud base (i.e., cloud base in the current grid box) and that the resulting droplet number tendency is proportional to the fractional area of the local cloud base (i.e., the $f_\mathrm{b}$ in Eq. 9). Figure 11d shows that, when ultra-low CDNCs are simulated (red line), the fractional area of local cloud base, $f_\mathrm{b}$, has composite mean values

very close to zero, which explains at least partially the lack of droplet nucleation in such cases: cloud-base nucleation cannot occur when there is no cloud base.

Section 2.3.3 also pointed out that, when there is neither droplet nucleation at the local cloud base nor nucleation associated with cloud fraction increase, there still can be, in principle, a source of droplet number caused by turbulent mixing. This potential source, however, also appears to be absent in the cases with ultra-low CDNCs (Fig. 11c, red line). We note that a

significant source of droplet number caused by turbulent mixing would require (I) substantial nucleation in other grid layers to cause vertical gradients and (II) sufficiently strong turbulent eddies to carry out down-gradient transport. The online conditional sampling performed in this study was pointwise (gridboxwise), as the current software infrastructure cannot easily capture column-wide information for each of the sampled grid boxes. Therefore we do not have good model output to reason about factor (I). As for factor (II), Fig.11e shows the composite mean of $w^*_\mathrm{act}$ defined in Eq. (2), and Fig.11f shows the $\sigma_w$ defined

in Eq. (3). While $w^*_\mathrm{act}$ is used for calculating the activated aerosol fractions and has an assumed minimum of $0.1 \, \mathrm{m \, s}^{-1}$ as in various other models (see, e.g., discussions in Poku et al., 2021), $\sigma_w$ does not have this assumed minimum and is an indicator of the strength of turbulence. In the cases of ultra-low CDNCs in Arctic July, the mean $\sigma_w$ is about $0.2 \, \mathrm{m \, s}^{-1}$ in the lowest model



layer but quickly drops to $0.04\,\mathrm{m\,s^{-1}}$ or less at about $950\,\mathrm{hPa}$ and higher altitudes (Fig. 11f), and the values are systematically and substantially smaller than those seen for the higher CDNC ranges (Fig. 11f). This means even though droplet nucleation

might happen in other layers of the grid column, turbulent transport from those layers is expected to be weak as turbulence is weak. Overall, it is understandable that when the simulated cloudy regions are relatively deep (i.e., cover multiple grid layers) and the cloud fractions vary little in the vertical as well as in time, the lack of local nucleation and the rather weak turbulence mixing can lead to a lack of droplet number sources in model layers away from the main cloud base.

Counterparts of Fig. 11 showing the results for January over the North Pacific storm track, March over SE China, and January

over SE United States can be found in Figs. S10–S12 in the Supplement. The focus months were chosen for their relatively high frequencies of occurrence of ultra-low CDNCs (see Fig. S6 in the Supplement). All the examined regions show the same qualitative features: compared to the cases with higher CDNCs, the ultra-low CDNCs are associated with negligible droplet nucleation, lack of local cloud base, weak turbulence, and negligible turbulent mixing. These results, especially the fact that all sources and sinks represented by the DROPMIXNUC parameterization are negligibly small when ultra-low CDNCs are found,

provide additional evidence to support the conclusion drawn at the end of Sect. 5.1 that the ultra-low CDNCs in EAMv2 are likely caused by weak sources rather than strong sinks.

### 5.3 Droplet nucleation problem or aerosol concentration problem?

At this point, one might start wondering whether the lack of cloud droplet number source identified in the cases of ultra-low CDNCs is caused by the lack of aerosol particles (APs), noting that if APs were abundant, even small activation fractions and

relatively weak turbulent transport might be sufficient to avoid ultra-low CDNCs. To help answer this question, we present in Fig. 12 the composite mean number concentrations of cloud condensate nuclei (CCN) at 0.1% supersaturation. The CCN number concentration (unit: $\mathrm{cm^{-3}}$) was diagnosed during the simulation using

$$\rho_{\mathrm{air}}\overline{N}_{\mathrm{CCN}} = \rho_{\mathrm{air}} \sum_{i=1}^{M} \left[ f_{a,i}^{*} \left( \overline{N}_{a,i} + \overline{N}_{c,i} \right) \right], \tag{11}$$

where $\rho_{\mathrm{air}}$ is air density, and $\overline{N}_{a,i}$ and $\overline{N}_{c,i}$ are the grid box mean interstitial and cloud-borne aerosol number mixing ratios,

respectively. The activation fractions, $f_{a,i}^{*}$, $i = 1, \cdots, M$, were calculated following Eqs. (13) and (15) in ARG2000, namely,

$$f_{a,i}^{*} = \frac{1}{2} \left[ 1 - \mathrm{erf}(u_i^*) \right] \tag{12}$$

with

$$u_i^* = \frac{2\ln(S_{mi}/S^*)}{3\sqrt{2}\ln\sigma_i}. \tag{13}$$

Here $S_{mi}$ is the critical supersaturation of APs with the geometric mean radius of aerosol mode $i$ (Eq. 9 in ARG2000 and

Eq. 8 in Abdul-Razzak et al., 1998). $\sigma_i$ is the geometric standard deviation of aerosol mode $i$ (Eq. 2 in ARG2000). $S^*$ is the prescribed supersaturation of 0.1% instead of the maximum supersaturation, $S_{\mathrm{max}}$, in ARG2000. In other words, the diagnosed CCN concentration is the number concentration of APs of all sizes and compositions in the grid box that would be activated at 0.1% supersaturation.





In Fig. 12, the composite mean CCN concentrations are shown for the three CDNC ranges and for all four regions discussed above. The thin black vertical line in each panel is a reference line corresponding to the number concentration of 10 cm$^{-3}$. For all regions and altitudes shown in the figure, the composite mean CCN concentrations associated with ultra-low CDNCs are higher than 10 cm$^{-3}$. This is especially so in SE China, where the mean CCN concentrations between 700 hPa and the surface are generally higher than 400 cm$^{-3}$. In other words, if all APs in the grid box were deemed available for growth into cloud droplets, and if the balance between the supersaturation production in the ambient atmosphere and the supersaturation consumption by the APs' size growth translated to an $S_{\max}$ of 0.1% in the ARG2000 aerosol activation parameterization, we would expect cloud droplet number concentrations about 2 orders of magnitude higher than the simulated values. It follows that the calculation of droplet nucleation (rather than the simulated AP concentration) plays a primary role in causing ultra-low CDNCs in this region and time of year.

The composite mean CCN concentrations associated with ultra-low CDNCs in the lower troposphere turn out to be in the range of about 20–50 cm$^{-3}$ in January over the North Pacific storm track, 30–40 cm$^{-3}$ in Arctic July, and 10–120 cm$^{-3}$ in January over SE United States. These concentrations are higher than 10 cm$^{-3}$ but not as dramatically as in SE China. Also noting that in these regions, the CCN concentrations associated with different CDNC ranges are significantly different (i.e., the two-standard-deviation ranges hardly overlap), we speculate the AP concentrations also play a role in causing ultra-low CDNCs in these regions. Future work evaluating both the CDNC and AP concentrations against observational data will be helpful. It will be useful to carry out a follow-up study to understand the reasons for the differences between the diagnosed CCN concentrations and the simulated CDNC concentrations, identifying which of these differences in various regions and in different CDNC ranges are physically justifiable and which ones are indication of the need for model improvement.

## 6 Additional experiments and discussions

The results in the previous section suggest that weak droplet nucleation is an important cause of ultra-low CDNCs in stratiform and shallow convective clouds in EAMv2. We now present some additional investigations and discussions along this line.

### 6.1 Motivation and design

Since the results presented above suggest that the sampled cases with ultra-low CDNCs and large cloud fractions feature weak turbulence, one might ask the following questions:

- How would the simulated CDNCs and global mean ERF$_{\mathrm{aer}}$ respond if stronger sub-grid updraft velocities were simulated by EAMv2 to cause stronger aerosol activation?

- How would the simulated CDNCs and global mean ERF$_{\mathrm{aer}}$ respond if stronger turbulence were simulated by EAMv2 to cause not only more aerosol activation but also stronger sub-grid transport of droplet number from cloud base? In other words, would it be worth adopting the assumption of vertically constant CDNC across contiguous cloudy grid boxes in a column, as done in some other models?





The sensitivity experiments nc00_w10 and nc00_w10k10 in Table 1 under group III were designed to provide qualitative answers to these questions. In the experiment nc00_w10, the $w_{\mathrm{act}}^*$ defined in Eq. (2) was artificially enhanced by a factor of 10 to affect aerosol activation but not turbulent mixing; in the experiment nc00_w10k10, both $w_{\mathrm{act}}^*$ and the turbulent diffusivity coefficient, denoted below as $\mathcal{K}_{\mathrm{mix}}$, were enhanced by a factor of 10. To avoid causing substantial drifts of the mean climate, these changes were applied

– only in the DROPMIXNUC parameterization (i.e., with no code changes in the turbulent transport of heat and moisture parameterized with CLUBB),

– only in grid layers with pressure higher than 600 hPa, and

– only in grid columns where three or more layers in that altitude range had $\sigma_w \leqslant 0.1~\mathrm{m\,s^{-1}}$ and $f_{\mathrm{c}} \geqslant 0.9$.

The results are shown in Fig. 13 and Fig. 3.

Before discussing the results from these sensitivity experiments, it is worth pointing out an additional feature of the atmospheric conditions associated with ultra-low CDNCs. In panel (g) of Fig. 11, the skewness of sub-grid vertical velocity calculated by CLUBB is shown. Compared to the cases with higher CDNCs, ultra-low CDNCs are associated with significantly smaller skewness, especially in grid layers near and below ∼900 hPa. The smaller skewness of sub-grid vertical velocity (Fig. 11g), weaker turbulence (Fig. 11f), larger cloud fractions (Fig. 10a), and lack of local cloud base in the grid

boxes (Fig. 11d), together with the ParaView screenshots shown in Fig. 8 and the animation in the video supplement, give the impression that the ultra-low CDNCs in Arctic July often occur in lower-level stratus clouds with large horizontal and vertical extents and weak turbulence. The higher CDNC values are associated with larger skewnesses of sub-grid vertical velocity (Fig. 11g), stronger turbulence (Fig. 11f), significant areas of local cloud base (Fig. 11d), and smaller cloud fractions (Fig. 10e), which suggests these cases are more convective. These qualitative features are also seen in the other three regions,

as shown in Figs. S10–S12 in the Supplement. To further analyze the atmospheric environment, it would be useful to examine more physical quantities in the grid columns, for example the surface fluxes, boundary layer stability, atmospheric conditions and aerosol concentrations at the main cloud base, etc. As mentioned earlier, the conditional sampling performed in this study was pointwise, hence we will defer the additional analyses to follow-on investigations.

## 6.2    Response of cloud droplet number

To present the responses of the simulated CDNCs to enhanced turbulence, we use here a metric $\widehat{N}_{l,\mathrm{top}}$ from a protocol of the AeroCom community, namely, the in-cloud CDNC at the top of liquid water clouds. The top of liquid water clouds in a grid column is defined as the highest-in-altitude model layer with grid box mean cloud liquid mass mixing ratio higher than $10^{-5}~\mathrm{g\,kg^{-1}}$. As instructed by bullet one under "Q/A" at https://wiki.met.no/aerocom/indirect, the time-averaged grid box mean CDNC and cloud fraction were diagnosed online and archived as model output. The annual mean $\widehat{N}_{l,\mathrm{top}}$ was calculated

by dividing the annual mean grid box mean liquid-cloud-top CDNC by the annual mean liquid-cloud-top cloud fraction.

The upper row in Fig. 13 presents the annual mean $\widehat{N}_{l,\mathrm{top}}$ simulated under the PD emission scenario. Panel (a) shows the geographical distribution simulated by the control model, i.e., the nc00 configuration; panels (b) and (c) show the changes caused





by enhancing $w_{\mathrm{act}}^*$ alone or enhancing both $w_{\mathrm{act}}^*$ and $\mathcal{K}_{\mathrm{mix}}$. Overall, the systematic changes in $\widehat{N}_{l,\mathrm{top}}$ are positive, and Fig. 13c shows larger increases than Fig. 13b (note that the contour levels are logarithmic), suggesting stronger aerosol activation and

stronger droplet mixing can both help increase the simulated cloud droplet number. The largest systematic changes in $\widehat{N}_{l,\mathrm{top}}$ are seen in the Arctic, SE China, SE United States, and over the North Pacific storm track, matching some of the geographical regions identified earlier in Sect. 4.2 with frequent occurrences of ultra-low CDNCs in EAMv2. This confirms that weak turbulence plays an important role in causing ultra-low CDNCs in the model. Furthermore, our analysis so far has lead to the impression that the atmospheric conditions associated with ultra-low CDNCs exhibit features of low-level stratus clouds, while

low-level stratus clouds are known to be commonplace in the above-mentioned regions, suggesting we have likely identified a cloud regime that is important for the occurrences of ultra-low CDNCs in the model.

Among the different geographical locations, the largest absolute changes in $\widehat{N}_{l,\mathrm{top}}$ are found to occur in SE China and SE United States. This can be understood from the equations and results presented earlier. In Sect. 2.3, it has been explained that the characteristic updraft velocity, $w_{\mathrm{act}}^*$, is used to calculate the fractions of interstitial aerosol particles being activated. From

Eqs. (7) and (9), we see that the droplet number tendencies caused by the activation of interstitial APs in mode $i$, denoted below by $\mathcal{U}_i$, can be expressed in an abstract form as

$$\mathcal{U}_i := \left( \frac{\partial \overline{N}_l}{\partial t} \right)_{\mathrm{nuc},i} = \mathcal{G}_i \left( w_{\mathrm{act}}^* \right) \overline{N}_{a,i} . \tag{14}$$

The function $\mathcal{G}_i$ depends explicitly on $w_{\mathrm{act}}^*$ (because of the activated fraction $f_{a,i}$) while $\overline{N}_{a,i}$ does not (at least to the first order). Hence, a change in this nucleation rate (denoted below by $\delta \mathcal{U}_i$) due to a change in $w_{\mathrm{act}}^*$ (denoted below by $\delta w_{\mathrm{act}}^*$) can be

estimated using a Taylor expansion, namely,

$$\delta \mathcal{U}_i \approx \left( \frac{\partial \mathcal{U}_i}{\partial w_{\mathrm{act}}^*} \right) \delta w_{\mathrm{act}}^* \approx \left( \frac{\partial \mathcal{G}_i}{\partial w_{\mathrm{act}}^*} \right) \left( \delta w_{\mathrm{act}}^* \right) \overline{N}_{a,i} , \tag{15}$$

which is proportional to the interstitial AP concentration $\overline{N}_{a,i}$. Figure 12 has shown that the ultra-low CDNCs in the polluted regions occur under higher ambient AP concentrations than in remote regions. Hence, given the same increases in $w_{\mathrm{act}}^*$ (and the consequent increases in the activation fractions), the regions with higher AP concentrations are expected to feature larger

changes in the number of nucleated droplets.

## 6.3  Responses of ERF$_{\mathrm{aer}}$

The lower row of Fig. 13 presents the PD–PI differences in $\widehat{N}_{l,\mathrm{top}}$, again with the left panel showing results from the nc00 configuration and the middle and right panels showing changes caused by increasing $w_{\mathrm{act}}^*$ or both $w_{\mathrm{act}}^*$ and $\mathcal{K}_{\mathrm{mix}}$. The key signals seen here are increases—not only in $\widehat{N}_{l,\mathrm{top}}$ itself as seen earlier in the first row of the figure, but also in the PD–PI

differences of $\widehat{N}_{l,\mathrm{top}}$, suggesting there is a differential impact on $\widehat{N}_{l,\mathrm{top}}$ under PD and PI emission scenarios. The reason for the differential impact can again be explained with Eq. (15): the PD emissions lead to higher $\overline{N}_{a,i}$ than the PI emissions and hence larger $\delta \mathcal{U}_i$ (nucleation rate increase) in response to the same $\delta w_{\mathrm{act}}^*$. Keeping in mind the larger PI-to-PD increases in $\widehat{N}_{l,\mathrm{top}}$ in the sensitivity experiments nc00_w10 and nc00_w10k10, it is not surprising to see in Fig. 3 that the global mean annual mean ERF$_{\mathrm{aer}}$ is also stronger in magnitude (more negative) in these experiments.





Recall that this study was motivated by the commonly observed model behavior that using a lower bound to remove extremely low CDNCs can help weaken the simulated global mean $\mathrm{ERF_{aer}}$ in aerosol-climate models. In the sensitivity experiments discussed in this section, CDNCs are increased in regions where ultra-low values are found in the nc00 configuration, hence, we expect ultra-low CDNCs to occur less frequently; the corresponding $\mathrm{ERF_{aer}}$, however, strengthens rather than weakens in its magnitude. We conducted additional experiments in which the assumed minimum value of the characteristic sub-grid

updraft velocity, i.e., the $w_{\mathrm{act,min}}$ in Eq. (2), was changed. Those experiments exhibited the same qualitative behavior, namely, higher overall CDNCs are correlated with stronger (more negative) $\mathrm{ERF_{aer}}$. This qualitative behavior has been observed in other models (e.g., Gantt et al., 2014) and most recently in Ghosh et al. (2024). These results are undesirable from the $\mathrm{ERF_{aer}}$ perspective, but they do not contradict the weaker (less negative) $\mathrm{ERF_{aer}}$ seen in global models when lower bounds are applied to CDNC. At the locations and time steps where ultra-low CDNCs are simulated under both PD and PI emission scenarios, an

imposed lower bound will artificially eliminate the PD–PI differences in CDNC and consequently eliminate their contribution to $\mathrm{ERF_{aer}}$. We speculate this is how the use of $\mathrm{CDNC_{min}}$ reduces $\mathrm{ERF_{aer}}$ in EAMv2—and possibly in some other global models, as Mignot et al. (2021) have discussed for the IPSL-CM6A-LR model in Sect. 4.5 therein.

    It is also worth noting that our results presented here does not contradict the widely established notion of the cloud albedo susceptibility being weaker (less negative) in regimes with higher aerosol load and higher droplet number concentrations.

Figure 14 explains this point using a schematic in the spirit of Fig. 3 of Carslaw et al. (2013). The independent variable $x$ and dependent variable $y$ in Fig. 14 here can be interpreted as any of the pairs of variables depicted in panels (b)–(d) in Fig. 3 of Carslaw et al. (2013), although for our discussion here, the most pertinent pairs are (I) CDNC as $y$ and CCN concentration as $x$, and (II) cloud albedo as $y$ and anthropogenic aerosol emissions as $x$. All three curves in Fig. 14 here, $y_1$, $y_2$, and $y_3$, have the same qualitative feature as in Fig. 3 of Carslaw et al. (2013), namely, their slopes $(dy/dx)$ become less steep as $x$

increases. The $y_1$ curve schematically represents the nc00 configuration of EAMv2, and the $y_2$ curve depicts our sensitivity experiments with enhanced $w_{\mathrm{act}}^*$ (or $w_{\mathrm{act}}^*$ as well as $\mathcal{K}_{\mathrm{mix}}$). Taking the variable pair (I) stated above as an example, the $\Delta x$ in our schematic is an emission-induced CCN concentration change; $\Delta y_1$ and $\Delta y_2$ are the resulting droplet number changes in the nc00 configuration and in the sensitivity experiments, respectively. Figure 13 has shown that with a larger $w_{\mathrm{act}}^*$, both $\widehat{N}_{l,\mathrm{top}}$ and its PD–PI difference increase systematically in many geographical regions. For Fig. 14, this means the $y_2$ curve lies

above the $y_1$ curve and has steeper slopes than the $y_1$ curve in the depicted range $\Delta x$. The $y_3$ curve symbolizes a hypothetical model configuration that simulates higher droplet concentrations than both nc00 and our sensitivity experiments with enhanced turbulence—so much higher, so that for the CCN concentration range indicated by $\Delta x$, the simulation is much closer to the regime noted as "droplet formation limited by supersaturation" in Fig. 3b of Carslaw et al. (2013), and hence the $y_3$ curve has significantly gentler slopes that are similar to what is found on the $y_1$ and $y_2$ curves at much larger $x$ values. The schematic

illustrates that if we focus on a single curve (which corresponds to a single model configuration with a specific choice of model formulation and parameters for cloud droplet nucleation), $y$ is indeed less sensitive to $x$ in regimes with higher $x$ (and higher $y$, since the function is monotonic). However, different model configurations may appear as different curves in the schematic, and larger $y$ values do not necessarily correspond to smaller $dy/dx$ slopes across different curves. Hence, a model capable of



simulating higher CDNCs does not necessarily produce weaker ERF$_{aer}$. This point can, in fact, be inferred from the uncertainty
envelope diagram in Fig. 3e of Carslaw et al. (2013).

Results from both the literature and our experiments indicate that, while ultra-low CDNCs and strong ERF$_{aer}$ are related, the
relationship is complex. It is possible that the frequent occurrences of ultra-low CDNCs and relatively strong ERF$_{aer}$ in EAMv2
have the same root causes, but very likely, there are also various more direct factors that affect CDNC and ERF$_{aer}$ separately.
Before the root causes are addressed and the more direct impactors are well understood, the alleviation of one symptom might
not automatically lead to the mitigation of the other.

## 7   Summary and conclusions

This study aims at identifying where ultra-low CDNCs (i.e., in-cloud cloud droplet number concentrations lower than 10
cm$^{-3}$) occur in the stratiform and shallow convective clouds simulated by E3SMv2 and which of the identified situations have
the strongest impact on the simulated effective radiative effect of anthropogenic aerosols, ERF$_{aer}$. Process-level analyses are
conducted to reveal characteristics of the cloud droplet formation, transport, and removal processes associated with impactful
ultra-low CDNCs in stratiform and shallow convective clouds.

Simulations performed with present-day emissions show that ultra-low CDNCs occur most frequently over the mid- and
high-latitude oceans in both hemispheres, with a general trend of higher frequencies at higher latitudes (Fig. 5a-c). Over the
North Pacific storm track and in the Arctic region, among all the model time steps in a year and among all grid boxes in the
middle and lower troposphere, about 20–40% of the cases with cloud liquid presence are associated with ultra-low CDNCs
(Fig. 6a-b). In polluted continental regions like SE China and SE United States, this percentage is about 15% or higher (Fig. 6c-
d) despite the generally high aerosol concentrations (Fig. 12c-d).

Sensitivity experiments reveal that ultra-low CDNCs in the lower troposphere (with air pressure higher than 600 hPa) play the
primary role in determining the sensitivity of E3SMv2's global mean ERF$_{aer}$ to the lower bound of CDNC (CDNC$_{min}$ = 10 cm$^{-3}$,
Fig. 3). The NH middle latitudes and the Arctic region show the largest absolute changes of annual mean ERF$_{aer}$ when the lower
bound is imposed (Fig. 4b), while the Arctic region features the largest relative changes in the annual mean ERF$_{aer}$ (Fig. 4c).
In the Arctic, the strongest seasonal responses of ERF$_{aer}$ are found in boreal summer (Fig. 4f).

Ultra-low CDNCs are typically found in grid boxes with either very large (> 0.9) or very small (< 0.1) cloud fractions
(Fig. 7), and the former category is the larger contributor to the global mean ERF$_{aer}$'s sensitivity to CDNC$_{min}$ (Fig. 3). The
ultra-low CDNCs associated with large cloud fractions are typically found in contiguous grid boxes in the main body of large
cloudy regions (Figs. 8–9 and video supplement). These cases feature strong water vapor condensation (Fig. 10g, red line)
and weak turbulence (Fig. 11f, red line), while the statistical distributions of sub-grid vertical velocity feature relatively small
skewness (Fig. 11g, red line), suggesting the clouds are likely low-level stratus.

A comparison of the cloud droplet number budgets associated with different CDNC ranges suggests that the ultra-low
CDNCs are caused by weak sources of droplet number (Fig. 10c), especially lack of aerosol activation (Fig. 11a and b), rather
than strong sinks of droplet number (Fig. 10b, Fig. 10d, Fig. 11c).



CCN number concentrations were diagnosed by calculating the number concentrations of aerosol particles of all sizes and compositions in the grid box that would be activated at 0.1% supersaturation. In March over SE China, the composite mean CCN concentrations associated with ultra-low CDNCs are on the order of several hundred per $cm^3$ (Fig. 12c), suggesting
the simulated cloud droplet nucleation process (rather than the aerosol particle number concentration) plays the primary role in causing the ultra-low CDNCs in that region and time of year. In the selected focus months over the North Pacific storm track, the Arctic region, and SE United States, the composite mean CCN number concentrations associated with ultra-low CDNCs range from close to 20 $cm^{-3}$ to about 120 $cm^{-3}$ in the lower troposphere, and the values are significantly lower than the composite mean CCN concentrations associated with higher CDNCs (Fig. 12a, b, d), suggesting that the simulated cloud
droplet nucleation process and aerosol concentrations are both possible contributors to the ultra-low CDNCs.

Sensitivity experiments show that in lower-tropospheric grid boxes with large cloud fractions and weak turbulence, boosting the characteristic updraft velocity used in the aerosol activation parameterization or enhancing the turbulent mixing of cloud droplet number can increase the simulated CDNCs (Fig. 13b–c) in regions where ultra-low CDNCs are frequently found in EAMv2 (Fig. 5a–c) and where the use of $CDNC_{min}$ significantly affects $ERF_{aer}$ (Fig. 4b), suggesting the identified atmospheric
conditions are relevant. However, those increased CDNCs are accompanied by increased (rather than decreased) PD–PI differences in CDNC (Fig. 13e–f) as well as larger (rather than smaller) magnitudes of the global mean $ERF_{aer}$ (Fig. 3). This result is qualitatively consistent with findings from earlier studies such as Gantt et al. (2014) and most recently from Ghosh et al. (2024). Our interpretation of this undesirable correlation between changes in CDNC and changes in $ERF_{aer}$ is that, while the frequent occurrences of ultra-low CDNCs and relatively strong $ERF_{aer}$ in EAMv2 may have the same root causes, there are
likely more direct factors that affect CDNC and $ERF_{aer}$ separately. Here, we refrain from speculating what the root causes and more direct factors might be, noting that the E3SM model developers are actively working in this area (Shan et al., 2024a, b; Burrows et al., 2024). Nevertheless, our results presented in this paper clearly suggest that mid- and high-latitude low-level stratus occurring under weak turbulence is a cloud regime worth further investigating in the next steps.

*Code and data availability.* The E3SMv2 model codes and modifications as well as simulation and analysis scripts used in this paper can
be found in the Zenodo record 14517254 (Wan and Zhang, 2024, DOI:10.5281/zenodo.14517254). Data files containing output from the free-running simulations can be found in Wan (2024, DOI:10.25584/2481237). Data files containing results from the nudged simulations can be found in the Zenodo record 14518205 (Zhang and Wan, 2024, DOI:10.5281/zenodo.14518205).

*Video supplement.* An animation showing vertical cross-sections of Arctic summer clouds in the style of Fig. 8 can be found in the Zenodo record 14523176 (Yenpure et al., 2024a, DOI:10.5281/zenodo.14523176). An animation showing 3D visualizations of clouds in SE China
in the style of Fig. 9b–c can be found in the Zenodo record 14523263 (Yenpure et al., 2024b, DOI:10.5281/zenodo.14523263).



*Author contributions.* HW initiated the study, designed the numerical experiments, and performed and analyzed the free-running simulations. KZ performed and processed the nudged simulations. RCE, PJR, and XZ contributed to the design of analysis strategies and the interpretation of the results. AY and BG developed plugins for ParaView to support interactive exploration of EAMv2 simulation output. AY created the ParaView screenshots and animations with input from HW and BG. HW wrote the manuscript. All authors contributed to the revisions.

*Competing interests.* The authors declare that no competing interests are present.

*Acknowledgements.* The authors thank Harri Kokkola (Finnish Meteorological Institute), Daniel Grosvenor (University of Leeds and UK Met Office), and Steven J. Ghan (retired from Pacific Northwest National Laboratory) for informative discussions on the parameterization of cloud droplet nucleation in various global models. Yun Qian (Pacific Northwest National Laboratory) is thanked for his encouragement during the preparation of the manuscript. Some of the colormaps used in the figures are from Scientific Colour Maps 8.0.1 by Crameri (2023).

Funding: This study was supported primarily by the U.S. Department of Energy (DOE), Office of Science, Scientific Discovery through Advanced Computing (SciDAC) program, via a partnership in Earth System Model Development between the Biological and Environmental Research (BER) and the Advanced Scientific Computing Research (ASCR). KZ was supported by DOE BER through the E3SM project and by Pacific Northwest National Laboratory (PNNL) through the Laboratory Directed Research and Development Program. The work used resources of the National Energy Research Scientific Computing Center (NERSC), a U.S. Department of Energy Office of Science

User Facility located at Lawrence Berkeley National Laboratory, operated under contract DE-AC02-05CH11231, using NERSC awards ASCR-ERCAP0025451, ASCR-ERCAP0028881, and BER-ERCAP0030805. Additional computing resources were provided by the Compy supercomputer funded by DOE BER and managed by PNNL Research Computing. PNNL is operated by Battelle Memorial Institute for the U.S. Department of Energy under contract DE-AC06-76RLO1830.



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





**Table 1.** EAMv2 simulations presented in the paper. "PD" and "PI" refer to the aerosol and precursor emission scenarios, with PD being the 2005–2014 average and PI being values representing the year 1850. The free-running simulations were climatological runs. The nudged simulations were constrained by winds from the ERA-Interim reanalysis. Further details can be found in Sect. 3.

| Group | Short name | Brief description | Nudged simulations with PD or PI emissions | Free-running simulations with PD emissions |
|---|---|---|---|---|
| I | nc00 | No lower bound for CDNC | 2005–2014 | 10 years |
| I | nc10 | $CDNC_{min} = 10$ cm$^{-3}$ (EAMv2 default) | 2005–2014 | - |
| II | nc10_600hPa | $CDNC_{min} = 10$ cm$^{-3}$ only between 600 hPa and the Earth's surface | 2011 | - |
| II | nc10_f0.9 | $CDNC_{min} = 10$ cm$^{-3}$ only when cloud fraction is larger than 0.9 | 2011 | - |
| III | nc00_w10 | No lower bound for CDNC; updraft velocity used in activation calculation enhanced by a factor of 10 in selected grid columns and layers. | 2011 | - |
| III | nc00_w10k10 | No lower bound for CDNC; both turbulent mixing and the updraft velocity used in activation calculation enhanced by a factor of 10 in selected grid columns and layers. | 2011 | - |




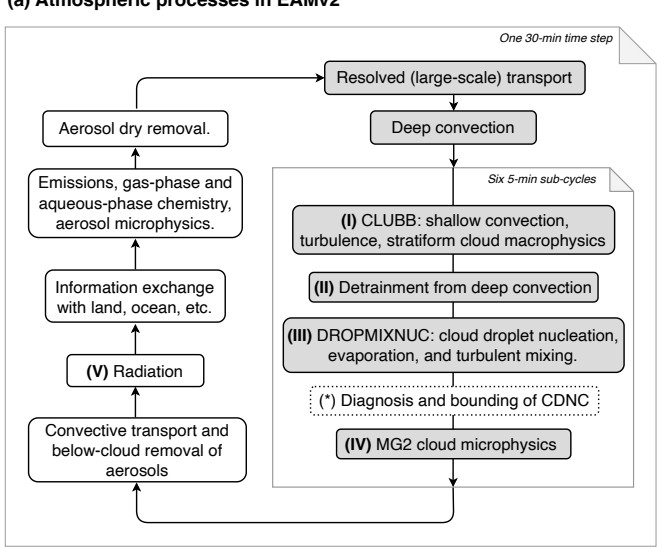

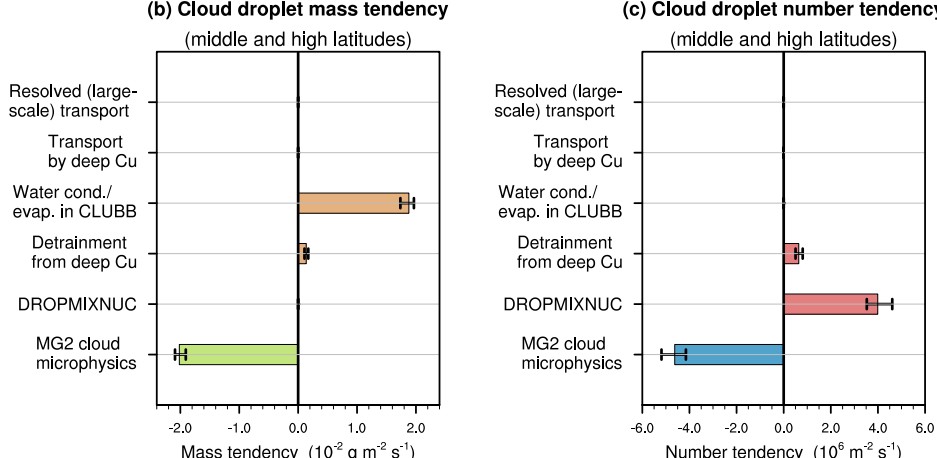

**Figure 1.** Panel **(a)** is a schematic showing the atmospheric processes considered in EAMv2 and the sequence of calculations during the model's time integration. Gray boxes correspond to processes that can directly change the mass or number mixing ratios of cloud droplets in stratiform and shallow convective clouds. Panels **(b)** and **(c)** show the time mean tendencies of the vertically integrated grid box mean droplet mass and number averaged over the middle and high latitudes. The filled boxes in panels **(b)** and **(c)** show annual averages, and the whiskers indicate the ranges of monthly averages. The budgets were diagnosed using one year of model output from a present-day EAMv2 simulation conducted without the prescribed lower bound of CDNC, i.e., configuration nc00 in Table 1.





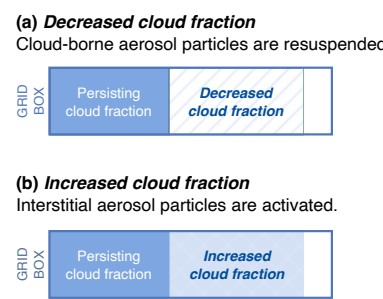

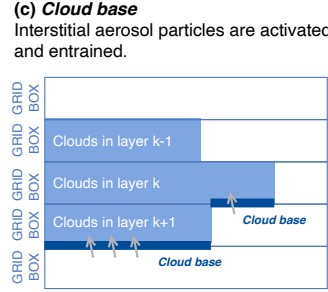

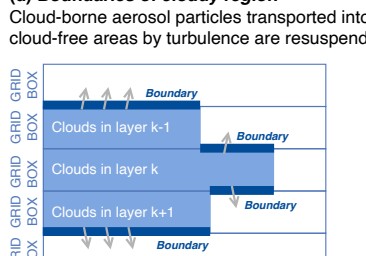

**Figure 2.** A schematic showing the treatment of different scenarios and physical processes in the parameterization of cloud droplet nucleation, evaporation, and turbulent mixing in EAMv2. More detailed descriptions can be found in Sect. 2.3.





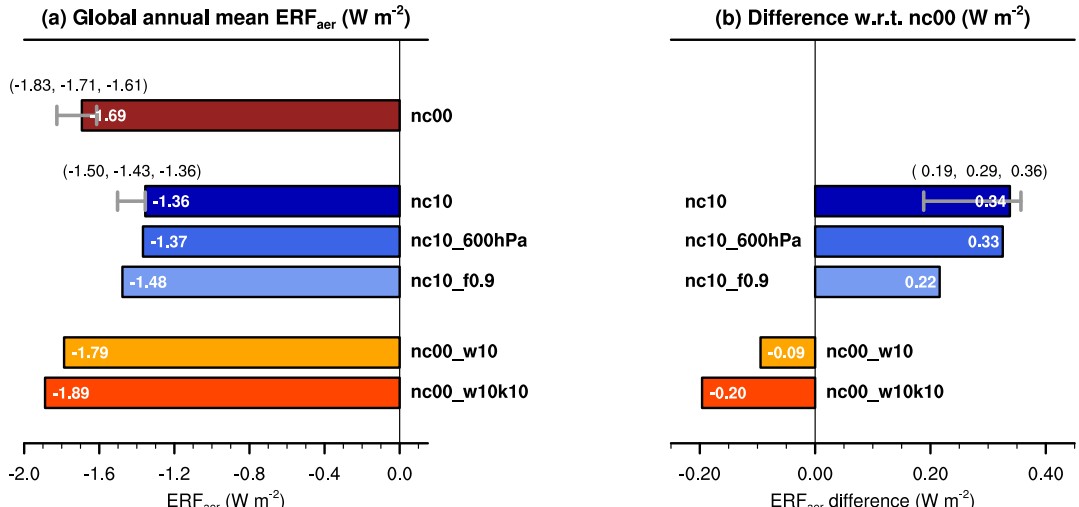

**Figure 3. (a)** Global annual mean ERF$_{aer}$ simulated with the various configurations of EAMv2 listed in Table 1. **(b)** Sensitivity of the global annual mean ERF$_{aer}$ to model configuration, shown as differences with respect to configuration nc00. In both panels, the filled rectangles and the numbers noted therein correspond to the model year of 2011. For the first two rectangles from the top in panel (a) and the first rectangle from the top in panel (b), the whiskers visualize the ranges of 1-year annual mean among the 10 simulations years, and the three numbers noted in each pair of parenthesis are the minimum, mean, and maximum values of the 1-year annual averages.





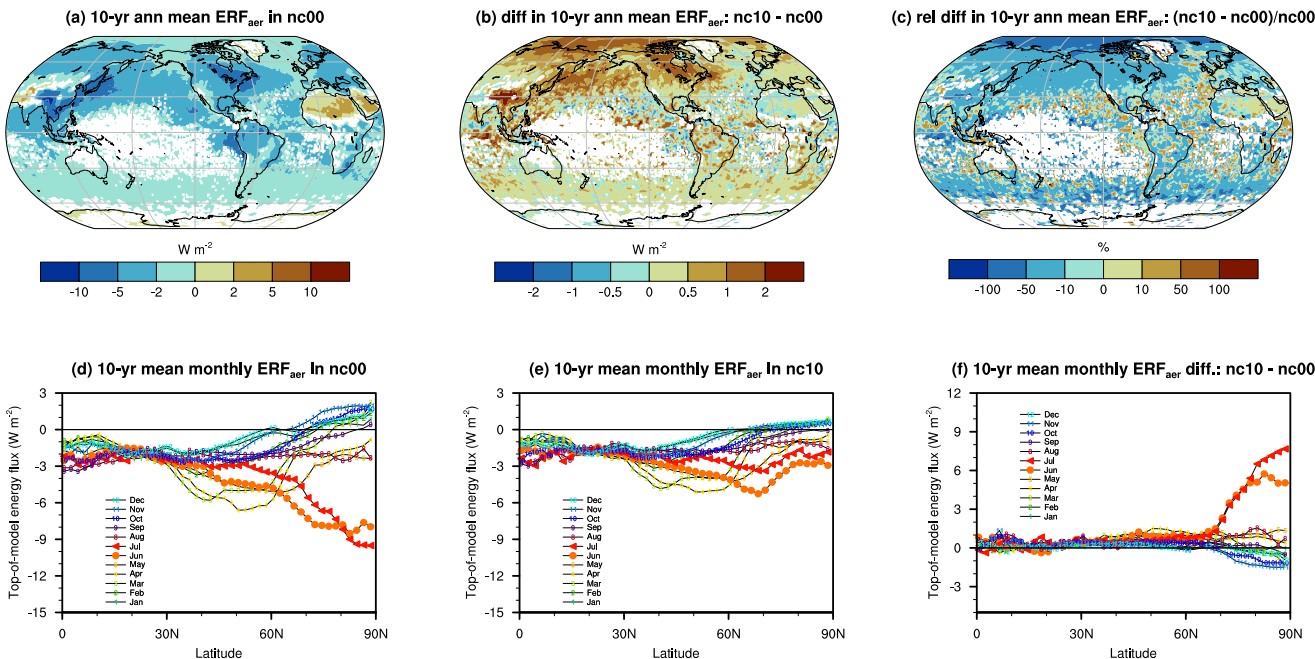

**Figure 4.** The upper row shows **(a)** the 10-year mean annually averaged $ERF_{aer}$ diagnosed from a pair of PD and PI simulations nudged to the meteorology of the years 2005–2014, conducted using EAMv2 with no lower bound for CDNC (i.e., configuration nc00), **(b)** changes in the 10-year annual mean $ERF_{aer}$ caused by imposing a lower bound of $CDNC_{min} = 10\ \mathrm{cm}^{-3}$ (configuration nc10 minus nc00), and **(c)** as in panel **(b)** but showing the relative changes. In these panels, masked out in white are the geographical locations where the $ERF_{aer}$ in configuration nc00 is statistically small (see Sect. 4.1). The lower row shows the 10-year mean zonal mean monthly $ERF_{aer}$ in the Northern Hemisphere: **(d)** in configuration nc00, **(e)** in configuration nc10, and **(f)** the differences. Similar figures showing the shortwave (SW) component of $ERF_{aer}$ and the SW component of the aerosol indirect effect (AIE) can be found in Figs. S2 and S3 in the Supplement. Further details and discussions can be found in Sect. 4.1.



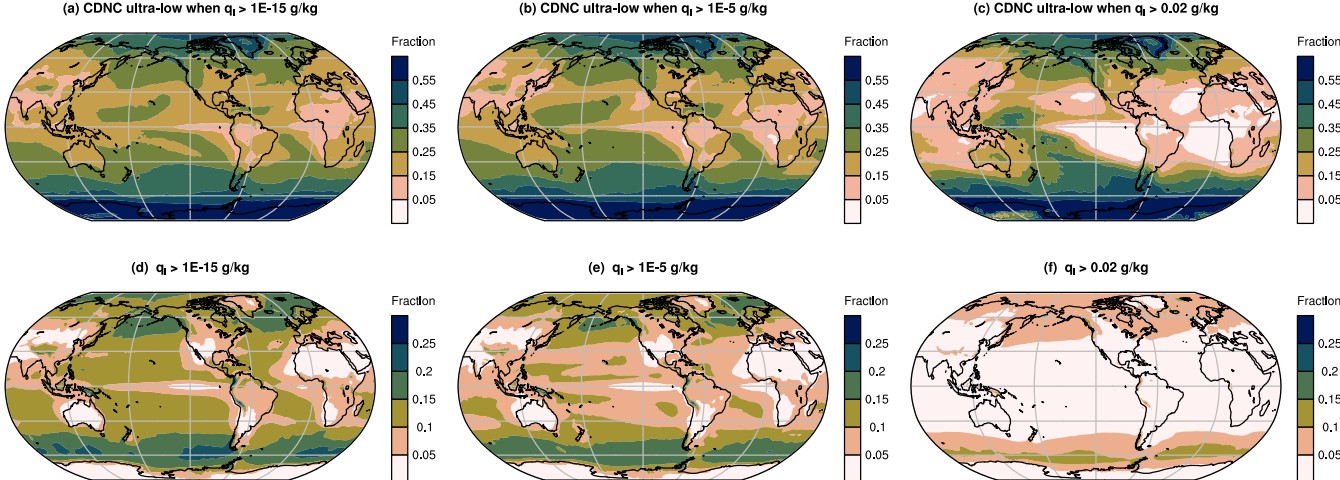

**Figure 5.** The upper row shows the 10-year mean annually averaged fraction of cloudy time steps and model layers that exhibit ultra-low CDNCs, i.e., the conditional probability $P\left(\text{CDNC ultra-low}\,\middle|\,\overline{q_l} \geqslant q_{l,\min}\right)$. The lower row shows the 10-year mean annually averaged fraction of all model time steps and model layers that are considered "cloudy" because the grid box mean cloud liquid mass mixing ratio of stratiform and shallow convective clouds, $\overline{q_l}$, is higher than a threshold $q_{l,\min}$, i.e., the probability $P\left(\overline{q_l} \geqslant q_{l,\min}\right)$. The three columns correspond to three different choices of $q_{l,\min}$ used in conditional sampling: $q_{l,\min} = 10^{-15}$ g kg$^{-1}$ in the left column, $q_{l,\min} = 10^{-5}$ g kg$^{-1}$ in the middle column, and $q_{l,\min} = 0.02$ g kg$^{-1}$ in the right column. The fractions (probabilities) were diagnosed in the 10-year free-running nc00 simulation explained in Table 1, i.e., EAMv2 with no lower bound for CDNC. Geographical distribution of the joint probability $P\left(\text{CDNC ultra-low} \cap \overline{q_l} \geqslant q_{l,\min}\right)$ can be found in Fig. S4 in the Supplement. Further details and discussions can be found in Sect. 4.2 .





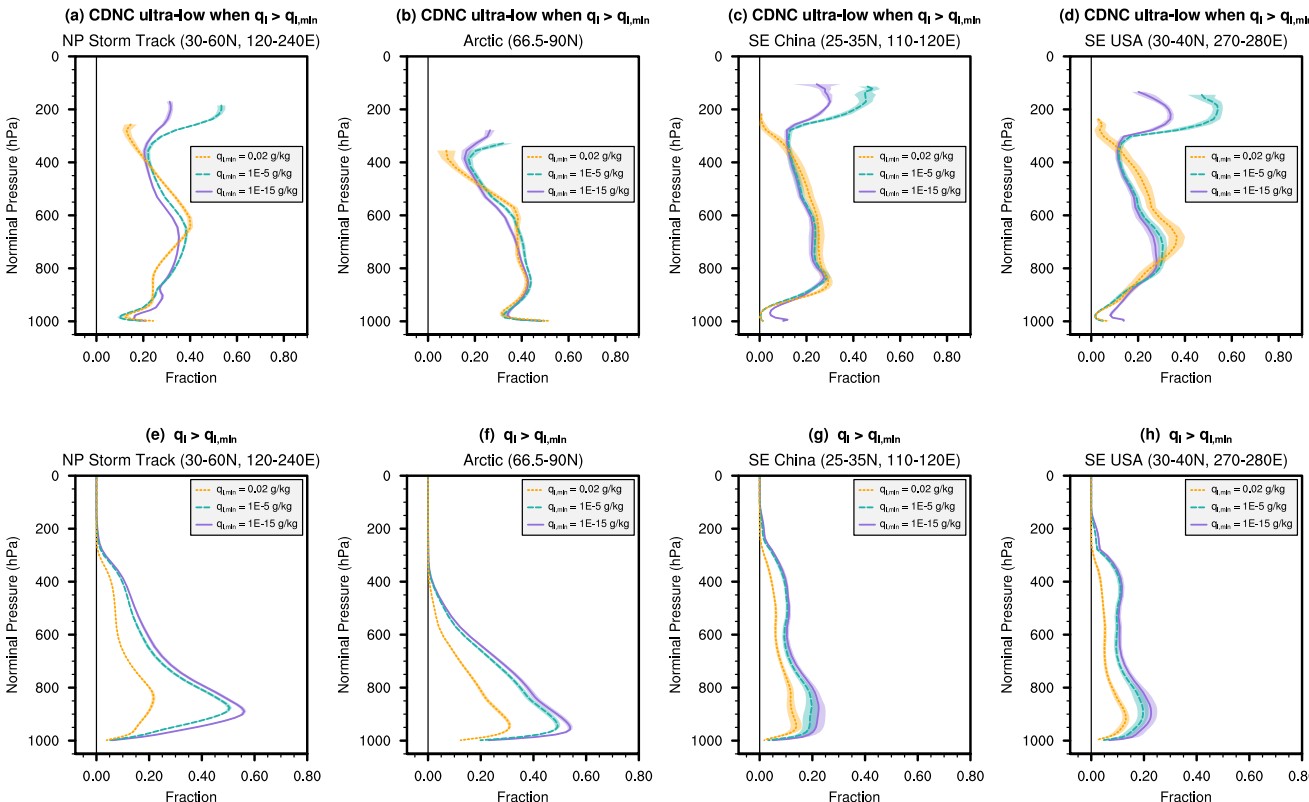

**Figure 6.** Regionally averaged vertical profiles of the annual mean fractional frequency of occurrence of various conditions. Like in Fig. 5, the upper row shows the conditional probability $P\left(\text{CDNC ultra-low}\,\middle|\,\overline{q_l} \geqslant q_{l,\min}\right)$, i.e., the fraction of cloudy cases that exhibit ultra-low CDNCs, and the lower row shows the probability $P\left(\overline{q_l} \geqslant q_{l,\min}\right)$, i.e., the fraction of model time steps and vertical layers that are considered "cloudy" because $\overline{q_l} \geqslant q_{l,\min}$. The four columns correspond to different regions; from left to right are the North Pacific storm track, the Arctic region, SE China, and SE United States, respectively. Lines are 10-year averages, and color shading shows two standard deviations of 10 one-year averages. The solid, dashed and dotted lines correspond to different $q_{l,\min}$ values used in conditional sampling. The fractions (probabilities) were diagnosed from the 10-year free-running nc00 simulation explained in Table 1 (i.e., EAMv2 with no lower bound for CDNC). Vertical profiles of the joint probability $P\left(\text{CDNC ultra-low} \cap \overline{q_l} \geqslant q_{l,\min}\right)$ can be found in Fig. S5 in the Supplement. Further details and discussions can be found in Sect. 4.3.







**Figure 7.** The upper row shows bivariate histograms of the grid box mean cloud liquid mass mixing ratio, $\overline{q_l}$, and cloud fraction, $f_c$, for the ultra-low CDNCs occurring in model layers with nominal pressure higher than 600 hPa. The lower row shows bivariate histograms of the in-cloud droplet mass concentration (CDMC) and $f_c$. The model output used in this analysis was 120 days of 6-hourly instantaneous values from January, April, July, and October (30 days per month) of the first simulation year after spin-up. Like in Fig. 6, results from different focus regions are shown in different columns, as indicated above each plot panel. Further details and discussions can be found in Sect. 4.4.



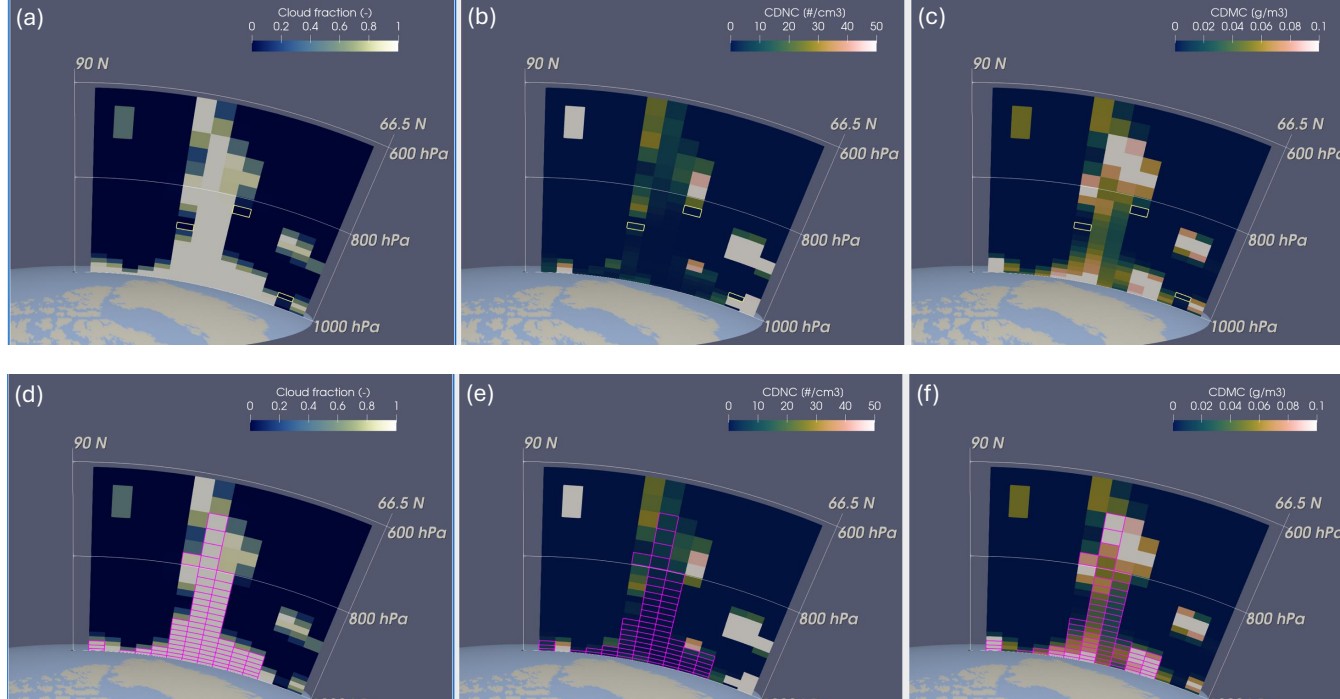

**Figure 8.** Vertical cross-sections of cloud fraction (left column), in-cloud CDNC (middle column), and in-cloud droplet mass concentration (right column) along the prime meridian northward of 66.5°N at 00UTC on July 3 of the first year in the free-running nc00 simulation. The color shading is the same in the upper and lower rows, but the overlaid quadrilateral frames in the upper row indicate model grid boxes associated with ultra-low CDNCs and *small* cloud fractions (<0.1), while the quadrilateral frames in the lower row indicate model grid boxes associated with ultra-low CDNCs and *large* cloud fractions (>0.9). An animation showing vertical cross-sections in the 30-day period from June 28 to July 27 can be found in the video supplement. See Sect. 4.4 for further details and discussions.





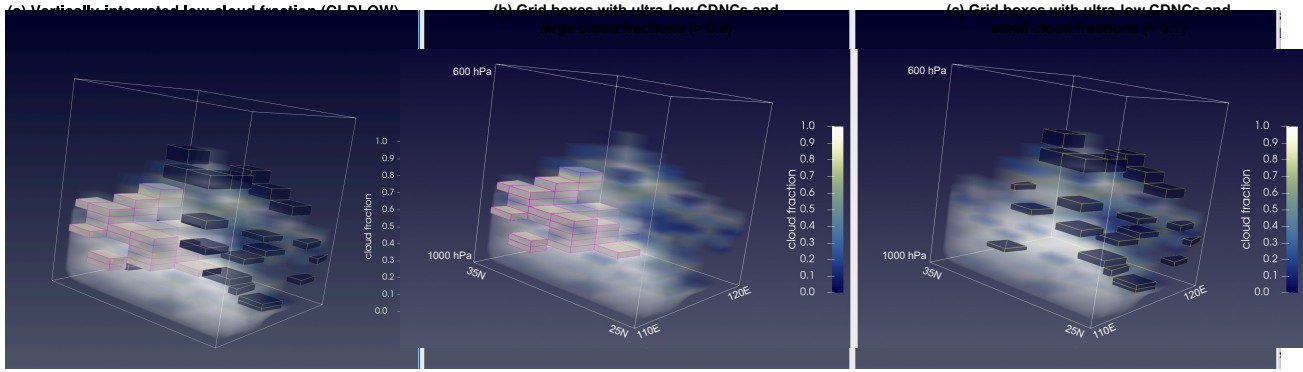

**Figure 9. (a)** Vertically integrated low-level cloud fraction at 00 UTC on March 8 of the first model year. The dashed square is the SE China region that panels (b) and (c) zoom into. **(b)** The semi-transparent color shading shows the cloud fraction values at the point of the model's time loop where the CDNCs are diagnosed and the lower bound is applied. The solid 3D boxes with magenta outlines are the grid boxes featuring ultra-low CDNCs and large cloud fractions (> 0.9). **(c)** As in panels (b), but the solid 3D grid boxes are associated with ultra-low CDNCs and small cloud fractions (< 0.1). An animation showing 3D visualizations in the 30-day period from February 28 to March 29 can be found in the video supplement. See Sect. 4.4 for further details and discussions.





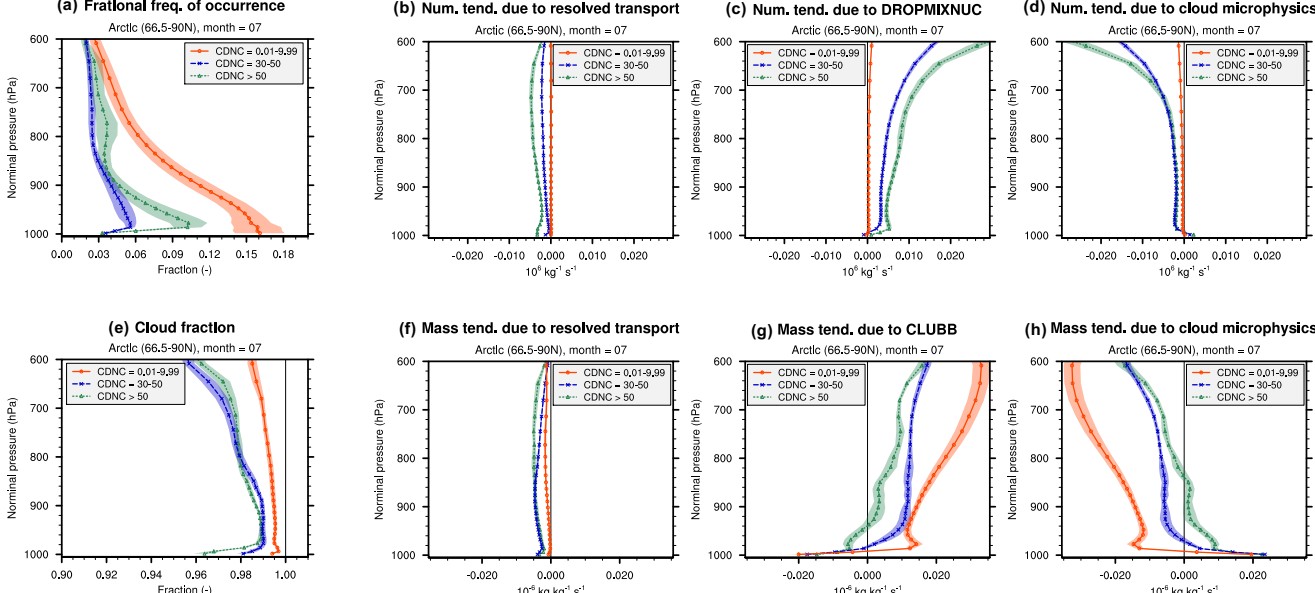

**Figure 10. (a)** Regionally averaged Arctic July mean frequency of occurrence of grid boxes and model time steps with cloud fraction larger than 0.9 and CDNC (unit: $cm^{-3}$) in three different ranges. **(b)–(d)** Composite mean of the grid box mean cloud droplet number tendencies caused by resolved transport, DROPMIXNUC, and cloud microphysics, respectively. **(e)** Composite mean cloud fraction. **(f)–(h)** Composite mean of the grid box mean cloud droplet mass tendencies caused by resolved transport, CLUBB, and cloud microphysics, respectively. Different marks and colors correspond to different CDNC ranges. The marks and lines show the averages of 10 Julys. Color shading indicates two standard deviations of the July average of individual years. Further details and discussions can be found at the beginning of Sect. 5 and in Sect. 5.1.



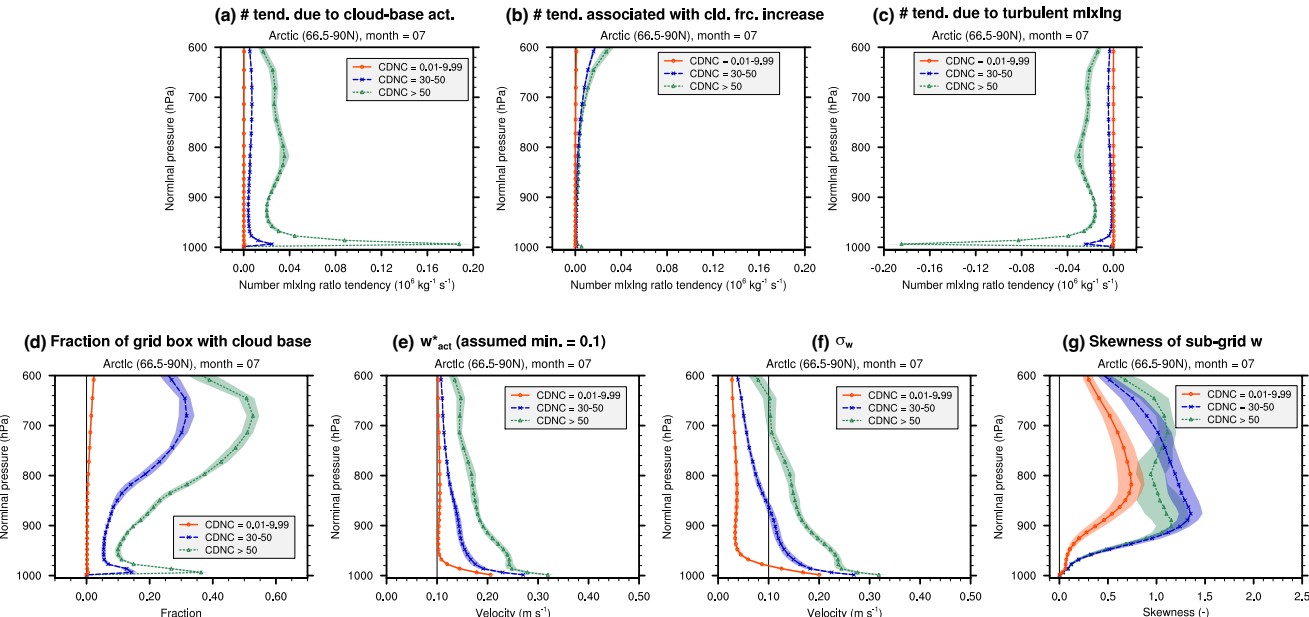

**Figure 11.** Regionally averaged composite mean of various quantities in Arctic July for three different CDNC ranges (CDNC unit: $cm^{-3}$). **(a)** Grid box mean droplet number mixing ratio tendency due to aerosol activation at the local cloud base in the grid box (Eq. 9 and Sect. 2.3.3). **(b)** Grid box mean droplet number mixing ratio tendency due to aerosol activation associated with cloud fraction increase (Eq. 7 and Sect. 2.3.2). **(c)** Grid box mean droplet number mixing ratio tendency due to turbulent mixing. **(d)** The fractional area of local cloud base in the grid box, i.e. the $f_b$ in Eq. (9) defined at the beginning of Sect. 2.3.3. **(e)** The characteristic updraft velocity used for calculating aerosol activation, see Eq. (2). **(f)** The sub-grid variance of vertical velocity, see Eq. (3). **(k)** Skewness of the sub-grid vertical velocity calculated by CLUBB. All results were derived from the 10-year free-running simulation using configuration nc00, i.e., EAMv2 with no lower bound for CDNC. The marks and lines show the averages of 10 Julys. Color shading indicates two standard deviations of the July average of individual years. Further details and discussions can be found in Sect. 5.2.

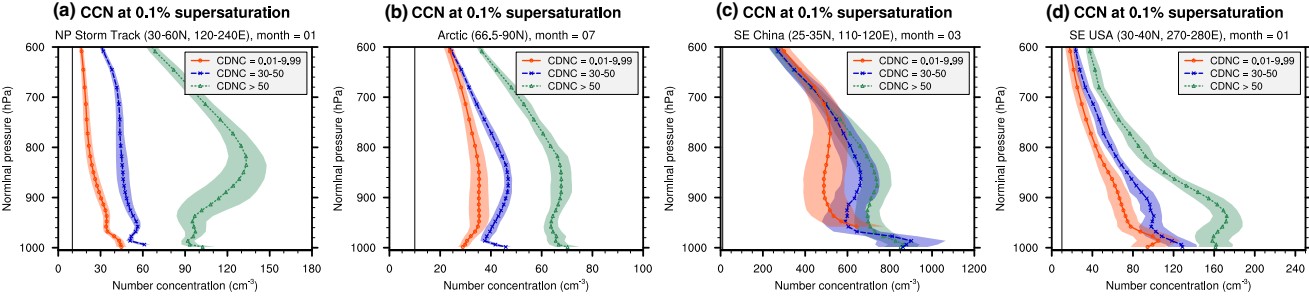

**Figure 12.** Regionally averaged monthly mean composite mean of the cloud condensation nuclei (CCN) number concentration at 0.1% supersaturation in different regions and focus months. From left to right: January over the North Pacific storm track, Arctic July, March over SE China, January over SE United States. The marks and lines show the averages of the focus month over 10 simulation years. Color shading indicates two standard deviations of the monthly average of different years. Further details can be found in Sect. 5.3.



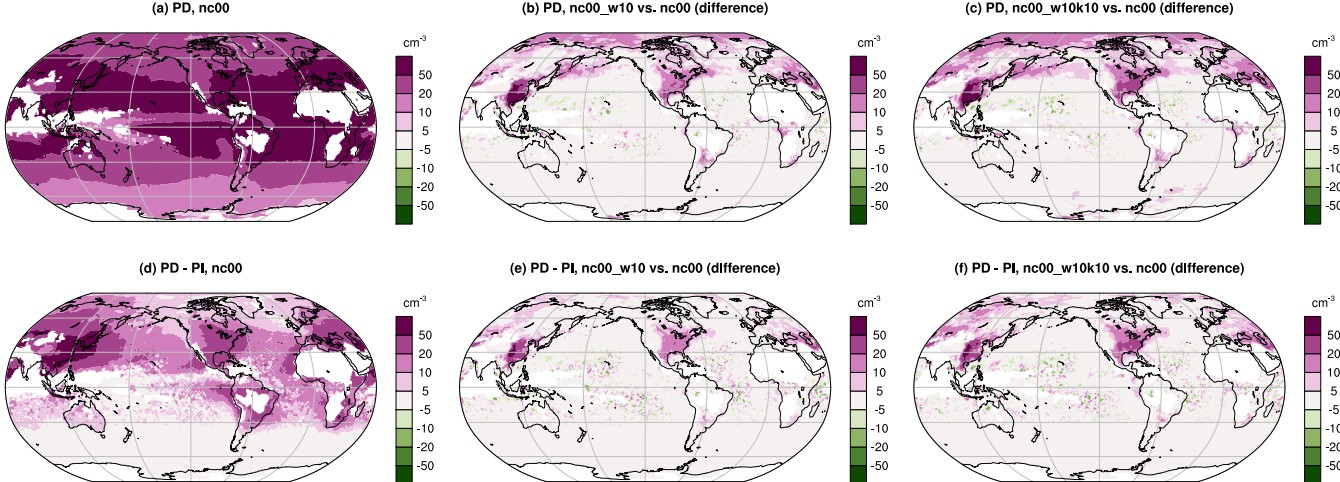

**Figure 13.** The upper row shows the annual mean in-cloud CDNC at the top of liquid water clouds, $\widehat{N}_{l,\text{top}}$, simulated under the PD emission scenario: **(a)** the results from the nc00 configuration, **(b)** the changes caused by enhancing the characteristic updraft velocity used in the aerosol activation calculation in the lower tropospheric grid boxes with large cloud fractions and weak turbulence (nc00_w10 minus nc00), and **(c)** the changes caused by enhancing both the characteristic updraft velocity and the diffusion coefficient for cloud droplet number ((nc00_w10k10 minus nc00). The lower row is arranged in the same way but shows the PD–PI differences in $\widehat{N}_{l,\text{top}}$. Masked out in white are locations with liquid cloud cover lower than 5%. Details of the experimental design can be found in Sect. 6.1. Discussions of the results can be found in Sects. 6.2 and 6.3.

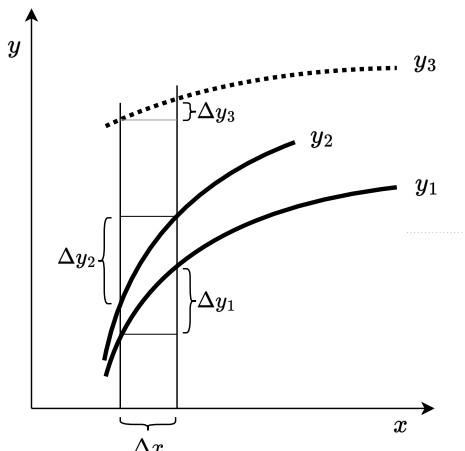

**Figure 14.** A schematic sketched in the spirit of Fig. 3 of Carslaw et al. (2013). The three curves, $y_1$, $y_2$, and $y_3$ symbolize three configurations of EAMv2 with differences in the parameters or model formulation used for cloud droplet nucleation in stratiform and shallow convective clouds. More details and discussions can be found in Sect. 6.3.