# Peer review of "Features of mid- and high-latitude low-level clouds and their relation to strong aerosol effects in the Energy Exascale Earth System Model version 2 (E3SMv2)"

_EGUsphere, 2024_

## Author Response (AR1)

**Point-by-point Response to Referee Comments**

Hui Wan, on behalf of all co-authors
March 7, 2025

(The referees' comments are shown in a bold italic font; authors' responses are in a regular font.)

**Referee 1**

*This paper investigates ultra-low (<10 cm$^{-3}$) cloud droplet number concentrations (CDNC) occurring in stratiform and convective clouds in E3SMv2, identifying the locations and some of the circumstances behind these events. These events are of particular concern because they appear to influence the global mean effective radiative forcing of anthropogenic aerosols (ERF$_{aer}$). Through a series of sensitivity experiments, the authors find that ultra-low CDNC events specifically in lower troposphere gridboxes with high cloud fractions and weak turbulence have a strong influence on ERF$_{aer}$ sensitivity. However, while enhancing activation and turbulent mixing in the simulations increases CDNC in these regimes, the magnitude of ERF$_{aer}$ increases simultaneously. Although this finding doesn't bring ERF$_{aer}$ closer to expected values, the results indicate that tropospheric stratus clouds in weak turbulence are a useful regime to focus on to help better understand the causal root behind these events.*

*Despite the many complex relationships and ideas discussed in this paper, the authors' precise prose and clean, frequently-referenced figures clearly explain the processes involved with minimal sources of confusion. It's a nicely put-together analysis that doesn't overwhelm the reader despite the high volume of figures in the paper and supplementary materials. In particular, their timely references to Figure 1 throughout helped the overall analysis stay grounded in a familiar, understandable framework. Before publication, I recommend a few minor corrections below:*

We are grateful for the referee's very positive feedback. Our responses to the questions and suggestions can be found below.

*Comments:*

1. *Line 231: What model levels are nudged? What's the relaxation time? More information about the nudging would be useful here.*

    We added more details about the nudging being used, as shown by the screenshot below.

230   model atmosphere.

These pairs of PD–PI simulations were nudged to the ERA-Interim reanalysis (Dee et al., 2011) to help distinguish signals of aerosols effects from noise caused by natural variability ( Kooperman et al., 2012). Horizontal winds in the altitude range of approximately 0.5 hPa to 850 hPa (i.e., model levels 5 to 58 out of a total of 72) were nudged with a relaxation time of 6 hours. The reanalysis data were read in every 6 hours and

235   linearly interpolated to each model timestep (Sun et al., 2019; Zhang et al., 2022b). Winds in the near-surface layers as well as air temperature and humidity were not nudged, so as to retain good consistency between the climatology of the nudged simulations and the climatology of the free-running simulations (Timmreck and Schulz, 2004; Zhang et al., 2014; Sun et al., 2019; Zhang et al., 2022b, a). Zhang et al. (2022a) showed that 1-year nudged E3SM simulations were sufficient for revealing key signals in the annually averaged global mean, zonal mean, and global patterns of $ERF_{aer}$. In this study, however, the sensitivity

240   of $ERF_{aer}$ to $CDNC_{min}$ (i.e., $\Delta ERF_{aer}$ caused by $\Delta CDNC_{min}$) is examined also in different seasons (see Fig. 4), which requires

2. *Lines 384-387: The nc10_f0.9 experiment can account for ~65% of the reduction in ERF$_{aer}$, indicating that the high cloud fraction regime plays a larger role in this change. Is the other ~35% wholly or mostly attributable to the small cloud fraction regime?*

Since the cases with cloud fraction lower than 0.1 appear to form the other dominant regime in terms of case count, one might expect this regime to be mostly responsible for the other ~35% of reduction in ERFaer. On the other hand, a very low cloud fraction means a very small weight for the contribution of the cloudy portion to the grid-box mean radiative effects, which gives us a reason to expect the small-cloud-fraction regime to be not very impactful. We have performed an additional pair of PD and PI simulations and confirmed the latter to be true, i.e., bounding CDNCs only when cloud fraction is lower than 0.1 leads to a very small change in the global annual mean ERFaer. It then follows that the aggregate effect of the cases with cloud fractions between 0.1 and 0.9 is non-negligible. We have not done much analysis for the "in-between" cases but note that this regime will be useful to examine in the future. A brief discussion has been added to the manuscript, as shown by the screenshot below.

390   Our sensitivity experiment nc10_f0.9 was designed to answer the first question. In this pair of PD and PI simulations, the lower bound $CDNC_{min}$ was applied only to grid boxes with cloud fractions higher than 0.9. The change in the year-2011 mean global mean $ERF_{aer}$  relative to nc00 amounts to a reduction of 0.22 $W\,m^{-2}$ in magnitude, explaining a major portion (about 65%) of the reduction of 0.34 $W\,m^{-2}$ in magnitude obtained by eliminating all ultra-low CDNCs (Fig. 3b). We also conducted an additional experiment bounding CDNCs only in grid boxes with cloud fractions lower than 0.1; the resulting

395   global mean $ERF_{aer}$ turned out to be very similar to that of the nc00 configuration. It then follows that the aggregate effect of the cases with cloud fractions between 0.1 and 0.9 is non-negligible despite the relatively low case counts in those cloud fraction bins. How the elimination of ultra-low CDNCs affects $ERF_{aer}$ and its various components in the high-cloud-fraction and medium-cloud-fraction regimes is an interesting topic for future investigations.

3. *In Section 5, you introduce Figure 10, which illuminates some of the involved processes for these Arctic summer low CDNC events. In this section and in this figure caption, it is difficult at first to follow which simulation (out of the many described in this paper) is being analyzed. While it is eventually mentioned in the caption for Figure 11, an in-text reminder that these are results from the free-running version of nc00 towards the beginning of this section would be useful to help keep the reader on track.*

Thanks for raising this point. We have added clarifications near the beginning of Section 5 and at the beginning of the caption of Figure 10, as shown by the screenshots below.

**5  Cloud droplet budget analyses**

Generally speaking, ultra-low CDNCs in stratiform and shallow convective clouds could be caused by weak sources or strong sinks of droplet number, or both. To help identify the culprits, we present in Sect. 5.1 a budget analysis for the main groups of droplet formation, transport, and removal processes considered in EAMv2, namely the dynamical core and parameterizations depicted by gray boxes in Fig. 1a and described in Sect. 2.1. After that, Sect. 5.2 zooms into the DROPMIXNUC parameterization to analyze the processes depicted in Fig. 2 and described in Sect. 2.3. All results presented in this section are based on the 10-year free-running simulation using the nc00 configuration, i.e., EAMv2 with no lower bound for CDNC.

To put the numbers associated with ultra-low CDNCs into context, two additional CDNC ranges, 30–50 $\mathrm{cm}^{-3}$ and higher than 50 $\mathrm{cm}^{-3}$, were sampled for comparison, all under the condition of cloud fraction being higher than 0.9. For clarification,

**Figure 10.** Results from the 10-year free-running simulation using the nc00 configuration, i.e., EAMv2 with no lower bound for CDNC. **(a)** Regionally averaged Arctic July mean frequency of occurrence of grid boxes and model time steps with cloud fraction larger than 0.9 and CDNC (unit: $\mathrm{cm}^{-3}$) in three different ranges. **(b)–(d)** Composite mean of the grid box mean cloud droplet number tendencies caused

*Typos:*
*Line 24: Should these units be in Wm$^{-2}$ (instead of Wm$^{-1}$)?*
*Line 152: "The decrement in the grid box mean droplet number mixing"- missing word?*

These have been corrected. Thanks again for the careful review.

**Referee 2**

*This study investigates ultra-low cloud droplet number concentrations in the E3SMv2 global climate model, focusing on their occurrence and impact on the effective radiative forcing of anthropogenic aerosols. The analysis identifies where ultra-low CDNCs occur in stratiform and shallow convective clouds and examines their influence on ERFaer. Process-level insights into cloud droplet formation and removal mechanisms are provided. The study suggests that increasing aerosol activation and enhancing turbulent mixing can raise CDNCs but may also undesirably amplify global mean ERFaer. The findings highlight the importance of further investigating mid- and high-latitude low-level stratus clouds under weak turbulence to address the root causes of ultra-low CDNCs and their strong influence on ERFaer in E3SMv2.*

*Overall I think it's a very well-written and info-dense manuscript and would support as-is publication, even though the style is a bit unconventional - L374 science questions in the middle of the paper, and L445 L512's question style titles, they did bring intrigue for the reader. Thanks for making it an interesting read.*

We sincerely thank Referee 2 for appreciating our work and for being open to our somewhat unconventional narrative. The style of the narrative reflects the curiosity that drove us throughout this study. Thanks again for the positive feedback.